# *FlexPrefill*: A Context-Aware Sparse Attention Mechanism for Efficient Long-Sequence Inference

**Xunhao Lai**[1*], **Jianqiao Lu**[2*], **Yao Luo**[3], **Yiyuan Ma**[3], **Xun Zhou**[3],
[1]Peking University    [2]The University of Hong Kong    [3] ByteDance Inc
laixunhao@pku.edu.cn, jqlu@cs.hku.hk
{luoyao.0, mayiyuan.unicorn, zhouxun}@bytedance.com

## ABSTRACT

Large language models (LLMs) encounter computational challenges during long-sequence inference, especially in the attention pre-filling phase, where the complexity grows quadratically with the prompt length. Previous efforts to mitigate these challenges have relied on fixed sparse attention patterns or identifying sparse attention patterns based on limited cases. However, these methods lacked the flexibility to efficiently adapt to varying input demands. In this paper, we introduce *FlexPrefill*, *a **Flex**ible sparse **Pre-fill**ing mechanism* that dynamically adjusts sparse attention patterns and computational budget in real-time to meet the specific requirements of each input and attention head. The flexibility of our method is demonstrated through two key innovations: 1) Query-Aware Sparse Pattern Determination: By measuring Jensen-Shannon divergence, this component adaptively switches between query-specific diverse attention patterns and predefined attention patterns. 2) Cumulative-Attention Based Index Selection: This component dynamically selects query-key indexes to be computed based on different attention patterns, ensuring the sum of attention scores meets a predefined threshold. *FlexPrefill* adaptively optimizes the sparse pattern and sparse ratio of each attention head based on the prompt, enhancing efficiency in long-sequence inference tasks. Experimental results show significant improvements in both speed and accuracy over prior methods, providing a more flexible and efficient solution for LLM inference.

 https://github.com/bytedance/FlexPrefill

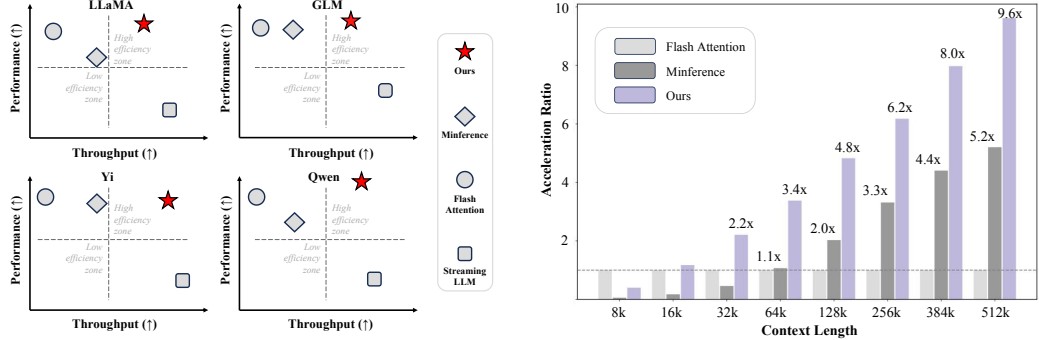

(a) Performance comparison of different inference approaches across LLaMA, GLM, Yi, and Qwen models.

(b) Attention acceleration ratio comparison of different approaches across various context lengths.

Figure 1: Performance and speed analysis of different inference approaches.

---

[*]Leading co-authors with equal contribution.

# 1 INTRODUCTION

Large language models (LLMs) are transforming various domains by enabling sophisticated natural language understanding and generation (Zhao et al., 2023). However, as the length of the context supported by these models increases (Zeng et al., 2024; Reid et al., 2024; Dubey et al., 2024; Abdin et al., 2024; Young et al., 2024), and as these models are applied to longer and more complex tasks (Bai et al., 2024; Shaham et al., 2023; An et al., 2024; Zhang et al., 2024c), the cost of inference, particularly during the prefill phase, becomes a bottleneck Fu (2024). The computational complexity of full attention mechanisms, which grows quadratically with sequence length, leads to inefficiencies during long-sequence inference tasks.

Considering the sparse nature of attention in LLMs (Deng et al., 2024), sparse attention mechanisms have been proposed to limit attention calculations to a selected subset of query-key pairs. Many existing sparse attention methods concentrate on predefined sparse patterns, such as BigBird (Zaheer et al., 2020), LongLora (Chen et al., 2024), and SlidingWindowAttention (Jiang et al., 2023). However, these approaches restrict the flexibility of sparse attention and often necessitate further training or fine-tuning of the model. There is also research proposing training-free sparse attention methods for large language models. For instance, StreamingLLM (Xiao et al., 2024b) identifies the attention sink phenomenon and introduces a sparse attention mechanism that combines initial tokens with sliding windows. Han et al. (2024) and Xiao et al. (2024a) further extend the sink attention approach. Additionally, other studies, such as those by Likhosherstov et al. (2021), Liu et al. (2021), and Jiang et al. (2024a), highlight the presence of dynamic sparse patterns in models. Minference (Jiang et al., 2024a) analyzes the intrinsic sparse patterns in LLMs and utilizes offline search-based sparse patterns along with online search-based sparse indexes. However, this approach relies on predefined sparse patterns and ratios with a limited search space, failing to account for the diverse and dynamic nature of real-world input sequences. Consequently, these attention patterns are often suboptimal, particularly when the complexity of input data varies, as they lack the adaptability required to meet the specific needs of varying input sequences.

To address these limitations, we propose *FlexPrefill*, a novel flexible sparse prefilling attention mechanism designed to adapt in real-time to the specific needs of each input and attention head. Our approach comprises two key components: **(1) Query-Aware Sparse Pattern Determination**: We categorize attention heads into *Diverse* pattern, which requires query-specific estimation, and *Structured* pattern, which is consistent across queries. Each attention head adaptively switches its sparse pattern based on the input by measuring the Jensen-Shannon divergence between the estimated and true attention score distribution. **(2) Cumulative-Attention Based Index Selection**: This component dynamically selects query-key indexes to be computed based on attention patterns, ensuring the sum of attention scores meets a predefined threshold. This approach allows for the adaptive allocation of computational budgets for different attention heads while maintaining the effectiveness of the model. *FlexPrefill*'s adaptive optimization of sparse patterns and ratios for each attention head based on inputs effectively balances computational efficiency with model performance.

We conduct extensive experiments using state-of-the-art LLMs, including Meta-Llama-3.1-8B-Instruct (Xiong et al., 2024), GLM-4-9B-Chat (Zeng et al., 2024), Yi-9B-200K (Young et al., 2024), and Qwen2-7B-Instruct (Yang et al., 2024), on challenging long-context benchmarks such as RULER (Hsieh et al., 2024) and InfiniteBench (Zhang et al., 2024c). The results demonstrate significant improvements in both speed and accuracy over prior methods, with *FlexPrefill* consistently preserving or even enhancing model performance across various context lengths and tasks.

# 2 SPARSE ATTENTION

**Problem Setting** In the sparse attention mechanism, $Q$ and $K$ represent the query and key matrices given sequence length $L$, respectively. The attention score for each position is computed as the scaled dot product between the query matrix $Q$ and the key matrix $K$, normalized by the square root of the head dimension $d$. To improve computational efficiency, sparse attention limits its computation to a subset of query-key pairs whose indexes form the set $S = \bigcup_{i=1}^{L} S_i$, where $S_i \subseteq \{(i,j) \mid 1 \le j \le i, j \in \mathbb{Z}\}$. The sparse attention mechanism is formalized as:

$$A(Q, K, V, S) = \text{Softmax}\left(\frac{1}{\sqrt{d}}\left(Q \cdot K^\top + M_S\right)\right) .$$

Here, $M_S$ is a sparse attention mask based on $S$, defined as:

$$M_S[i,j] = \begin{cases} 0, & \text{if } (i,j) \in S, \\ -\infty, & \text{otherwise.} \end{cases}$$

**Goal** The goal of the dynamic sparse attention system is to achieve a balance between computational efficiency and attention effectiveness. This can be formally expressed as a multi-objective optimization problem:

$$\min_{S} \quad |A(Q,K,V) - A(Q,K,V,S)|,$$
$$\min_{S} \quad |S|, \tag{1}$$

where $A(Q,K,V)$ represents the full attention result computed and $A(Q,K,V,S)$ represents the sparse attention result. The first objective in Equation 1 aims to minimize the difference between the sparse and full attention results. The second objective seeks to minimize the size of the selected subset $S$, which directly relates to the computational efficiency.

**Attention Sparse Patterns Exhibit Variability** In the attention mechanism of LLMs, we observe that attention patterns can vary significantly between attention heads. Figure 2a reveals a *Diverse* pattern for different query positions. This variability suggests that a one-size-fits-all approach to sparse attention may be suboptimal. Conversely, Figure 2b exhibits a more consistent and structured pattern across queries, showing a clear *Structured* pattern similar to that reported by Jiang et al. (2024a), which is also known as *Vertical-Slash* pattern. In attention heads with such a pattern, a subset of the attention map may be sufficient to estimate the entire sparse index set.

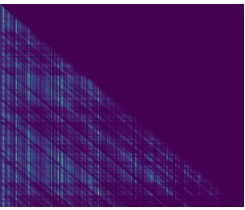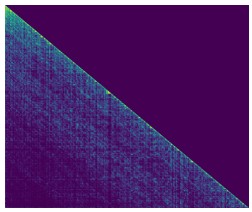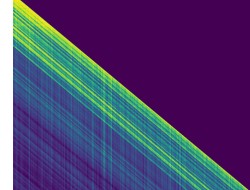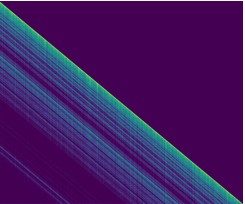

(a) Attention maps exhibiting *Diverse* patterns, where attended key tokens are scattered across different query positions with independent blocks.

(b) Attention maps exhibiting *Structured* patterns, e.g. attention is concentrated along vertical and slash-like structures, which is consistent across queries.

Figure 2: Comparison of attention patterns in different attention heads. The *Diverse* patterns (a) show scattered attention with independent blocks across query positions, while the *Structured* patterns (b) exhibit attention focused along certain structures.

**Varying Samples Require Adaptive Sparsity Ratio** Previous work (Jiang et al., 2024a) utilizes an offline search to determine a fixed sparsity ratio, applied uniformly across all input cases. However, as shown in Figure 3, the input prompts exhibit varying levels of complexity, necessitating different sparsity ratios for optimal performance. Samples with significant long-range dependencies may benefit from a lower sparsity ratio, whereas those with predominantly local dependencies can be efficiently processed with a higher ratio.

To address the above problems, we propose a method to dynamically adjust each sample's sparse pattern and sparsity ratio, as introduced in Section 3.

## 3 METHOD

In this section, we introduce our method for optimizing the sparse attention mechanism. Our approach involves two key components: (1) **Query-Aware Sparse Pattern Determination** and (2) **Cumulative-Attention Based Index Selection**. We then present the overall sparse attention algorithm of *FlexPrefill* in (3) **Algorithm**.

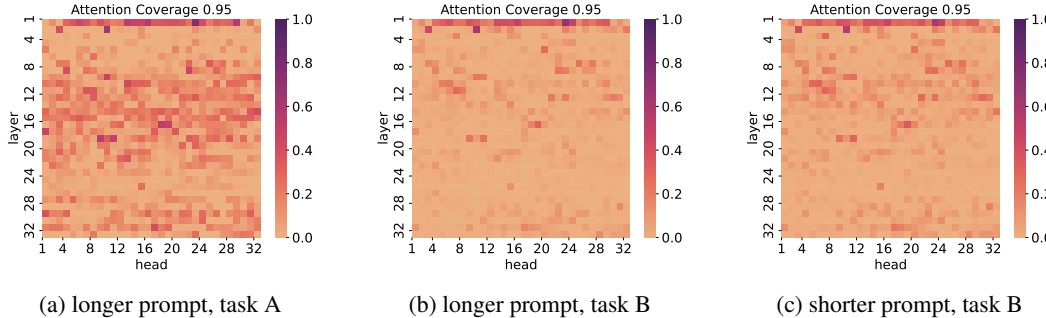

(a) longer prompt, task A     (b) longer prompt, task B     (c) shorter prompt, task B

Figure 3: Adaptive sparsity ratios across different attention heads and layers for varying sample complexities. Each heatmap shows the sparsity rate of different context lengths and task types given a fixed attention score coverage, where darker colors indicate more attention calculations. The sparsity distribution of different attention heads varies with different sample types (a, b) and different context lengths (b, c)

**Query-Aware Sparse Pattern Determination**    Our observations of *Diverse* (Figure 2a) and *Structured* patterns (Figure 2b) motivate an adaptive approach to sparse attention estimation. We classify attention heads into two types: the *Query-Aware* head, which is designed to capture variable sparse patterns contingent upon query positions, and the *Vertical-Slash* head, which is intended to handle commonly occurring structured sparse attention patterns in LLMs. We propose a method that dynamically determines whether to use *Query-Aware* attention estimates or a more consistent pattern, balancing computational efficiency with accuracy.

Given the computational challenges of calculating the full global attention map, we adopt a pragmatic approach:

1. **Representative Query Selection**: We select the last *block_size* query vectors, denoted as $\hat{\boldsymbol{Q}}$, as a representative subset for subsequent estimation.

2. **Block-Wise Attention Estimation**: By applying a pooling operation on the sequence, we compute two block-wise attention distributions:

$$\bar{\boldsymbol{a}} = \text{softmax}\left(\text{avgpool}(\hat{\boldsymbol{Q}})\text{avgpool}(\boldsymbol{K})^{\top}/\sqrt{d}\right),$$

$$\hat{\boldsymbol{a}} = \text{sumpool}\left(\text{softmax}\left(\hat{\boldsymbol{Q}}\boldsymbol{K}^{\top}/\sqrt{d}\right)\right),$$

where the kernel size of the pooling operation is equal to *block_size*, $\bar{\boldsymbol{a}}$ represents the estimated distribution and $\hat{\boldsymbol{a}}$ represents the true distribution.

3. **Distribution Comparison**: We quantify the discrepancy between these distributions using the square root of the Jensen-Shannon divergence:

$$D_{JS}(\bar{\boldsymbol{a}}, \hat{\boldsymbol{a}}) = \sqrt{JSD(\bar{\boldsymbol{a}}||\hat{\boldsymbol{a}})} = \sqrt{\frac{1}{2}\left(D_{KL}(\bar{\boldsymbol{a}}||\boldsymbol{m}) + D_{KL}(\hat{\boldsymbol{a}}||\boldsymbol{m})\right)}, \tag{2}$$

where $\boldsymbol{m} = \frac{1}{2}(\bar{\boldsymbol{a}} + \hat{\boldsymbol{a}})$ is the mean distribution, and $D_{KL}(\cdot||\cdot)$ denotes the Kullback-Leibler divergence. We adopt the Jensen-Shannon divergence for its symmetry and boundedness properties, which ensure a more stable and reliable pattern determination.

4. **Adaptive Decision**: We compare $D_{JS}$ to a predefined threshold $\tau$. If $D_{JS} < \tau$, the block-wise attention estimation adequately approximates the true distribution. Consequently, we calculate $\bar{\boldsymbol{a}}$ for all queries, enabling query-aware sparse index selection. Otherwise, the approximation is inadequate, we fall back to a more conservative approach. Specifically, we utilize only a subset of queries for subsequent sparse index searching and extend to the global index set using a vertical-slash pattern.

This adaptive method allows us to leverage Query-Aware attention patterns when appropriate, potentially capturing more diverse patterns. Meanwhile, it provides a fallback mechanism to a more consistent Vertical-Slash pattern when the attention estimation is unreliable.

**Cumulative-Attention Based Index Selection**  After deciding whether an attention head belongs to *Query-Aware* or *Vertical-Slash*, we attempt to minimize the computation while ensuring effectiveness. Our sparse attention mechanism aims to select the smallest possible subsets $\boldsymbol{S}_i$ for each query position $i$ while ensuring that the sum of normalized attention scores within each subset meets a predefined cumulative attention threshold $\gamma$. This objective can be formally expressed as:

$$\min_{\boldsymbol{S}_i} \sum_{i=1}^{n} |\boldsymbol{S}_i| \quad \text{subject to} \quad \sum_{(i,j)\in\boldsymbol{S}_i} \frac{\exp(\boldsymbol{Q}_i \cdot \boldsymbol{K}_j^\top / \sqrt{d})}{\sum_{j'=1}^{i} \exp(\boldsymbol{Q}_i \cdot \boldsymbol{K}_{j'}^\top / \sqrt{d})} \geq \gamma, \forall i \in [n] , \qquad (3)$$

where $\boldsymbol{S} = \cup_i \boldsymbol{S}_i$ and $|\boldsymbol{S}_i|$ denotes the size of chosen index subset, $\boldsymbol{Q}_i$ and $\boldsymbol{K}_j$ are the query and key vectors at position $i$, respectively. The detailed rationale behind optimizing Equation (3) is provided in Appendix A.

To achieve this objective, we employ different strategies depending on the attention head type:

1. *Query-Aware* **head**:
   - Estimate the block-wise attention scores by first average-pooling the queries and keys, then computing the attention.
   - Sort the blocks in descending order and select blocks in order until the sum of their normalized attention scores exceeds the threshold $\gamma$, forming the sparse index set $\boldsymbol{S}$.

2. *Vertical-Slash* **head**:
   - Select a representative query subset $\hat{\boldsymbol{Q}}$ and compute attention scores, then calculate average scores for the vertical and slash lines.
   - Sort these lines and select vertical and slash lines in descending order until their cumulative normalized attention scores exceed $\gamma$.
   - Extend selected lines to the entire attention matrix, forming the sparse index set $\boldsymbol{S}$.

This adaptive approach tailors the selection process to the attention patterns of each attention head, allowing us to efficiently determine the sparse index set $\boldsymbol{S}$ for sparse attention computation. By doing so, we balance computational efficiency with attention accuracy, ensuring that we focus computation on the most relevant token interactions while maintaining a predefined level of cumulative attention scores.

**Algorithm**  Algorithm 1 presents the overall procedure of our proposed method for efficient sparse attention computation. The algorithm takes the query matrix $\boldsymbol{Q}$, key matrix $\boldsymbol{K}$, value matrix $\boldsymbol{V}$, sparse pattern threshold $\tau$ and cumulative attention threshold $\gamma$ as input. It is divided into the following three parts:

(i) **Sparse Pattern Determination**: Algorithm 2 determines whether to use *Query-Aware* pattern or fall back to the *Vertical-Slash* pattern for each attention head.

(ii) **Sparse Index Selection**: Based on the attention patterns obtained in (i) and the given cumulative attention threshold $\gamma$, the sparse index set $\boldsymbol{S}$ that needs to be computed for each attention head is obtained by Algorithm 4 (*Query-Aware*) or Algorithm 3 (*Vertical-Slash*).

(iii) **Sparse Attention Computation**: The algorithm performs sparse attention computation for each attention head using the obtained sparse index and returns the final attention result.

# 4 EXPERIMENT

## 4.1 SETTINGS

In this section, we provide a comprehensive overview of the models and datasets employed in our experiments to assess the performance and capabilities of various long-context language models.

---

*FlexPrefill*

---

**Algorithm 1** Sparse Attention

**Input:** $\boldsymbol{Q}, \boldsymbol{K}, \boldsymbol{V} \in \mathbb{R}^{S \times d_h}$ , $\tau \in [0,1]$, $\gamma \in (0,1)$

*# Determine the sparse pattern based on the threshold $\tau$*
$pattern \leftarrow$ Sparse Pattern Search $(\boldsymbol{Q}, \boldsymbol{K}, \tau)$

*# Decide the sparse index set $\boldsymbol{S}$ based on pattern and $\gamma$*
**if** $pattern ==$ query_specific **then**
$\quad \boldsymbol{S} \leftarrow$ Query Aware Index $(\boldsymbol{Q}, \boldsymbol{K}, \gamma)$
**else if** $pattern ==$ vertical_slash **then**
$\quad \boldsymbol{S} \leftarrow$ Vertical Slash Index $(\boldsymbol{Q}, \boldsymbol{K}, \gamma)$
**end if**

*# Compute the final sparse attention output*
$\boldsymbol{y} \leftarrow \boldsymbol{A}(\boldsymbol{Q}, \boldsymbol{K}, \boldsymbol{V}, \boldsymbol{S})$
return $\boldsymbol{y}$

---

**Algorithm 2** Sparse Pattern Search

**Input:** $\boldsymbol{Q}, \boldsymbol{K}, \tau$

*# Take a representative query subset*
select $\hat{\boldsymbol{Q}} = \boldsymbol{Q}_{[-block\_size:]}$

*# Compute estimated block-pooled attention $\bar{a}$ and true block-pooled attention $\hat{a}$*
$\bar{\boldsymbol{a}} \leftarrow \text{softmax}\left(\text{pool}(\hat{\boldsymbol{Q}})\text{pool}(\boldsymbol{K})^\top/\sqrt{d}\right)$
$\hat{\boldsymbol{a}} \leftarrow \text{pool}\left(\text{softmax}\left(\hat{\boldsymbol{Q}}\boldsymbol{K}^\top/\sqrt{d}\right)\right)$

*# Compute Jensen-Shannon divergence*
$d_{JS} \leftarrow \sqrt{JSD(\bar{\boldsymbol{a}}||\hat{\boldsymbol{a}})}$

*# Determine whether to use Query-Specific attention pattern*
**if** $d_{JS} < \tau$ **then**
$\quad pattern \leftarrow$ query_specific
**else**
$\quad pattern \leftarrow$ vertical_slash
**end if**
return $pattern$

---

**Algorithm 3** Vertical Slash Index Search

**Input:** $\boldsymbol{Q}, \boldsymbol{K}, \gamma$

*# Compute a subset of the full attention map*
$\hat{\boldsymbol{A}} \leftarrow \text{softmax}\left(\hat{\boldsymbol{Q}}\boldsymbol{K}^\top/\sqrt{d}\right), where\ \hat{\boldsymbol{Q}} \subset \boldsymbol{Q}$

*# Sum and normalize attention scores along the vertical and slash directions*
$\boldsymbol{a_v} \leftarrow \text{sum\_vertical}(\hat{\boldsymbol{A}})/\sum_{i,j} \hat{\boldsymbol{A}}[i,j]$
$\boldsymbol{a_s} \leftarrow \text{sum\_slash}(\hat{\boldsymbol{A}})/\sum_{i,j} \hat{\boldsymbol{A}}[i,j]$

*# Sort vertical and slash attention scores*
$\boldsymbol{I_v} \leftarrow \text{argsort}(\boldsymbol{a_v})$
$\boldsymbol{I_s} \leftarrow \text{argsort}(\boldsymbol{a_s})$

*# Obtain the minimum computational budget making the sum of the scores exceeds $\gamma$*
$K_v \leftarrow \min\{k : \sum_{i \in \boldsymbol{I_v}[1:k]} \boldsymbol{a_v}[i] \geq \gamma\}$
$K_s \leftarrow \min\{k : \sum_{i \in \boldsymbol{I_s}[1:k]} \boldsymbol{a_s}[i] \geq \gamma\}$

*# Select vertical and slash index*
$\boldsymbol{S_v} \leftarrow \boldsymbol{I_v}[1:K_v],\ \boldsymbol{S_s} \leftarrow \boldsymbol{I_s}[1:K_s]$
$\boldsymbol{S} \leftarrow \boldsymbol{S_v} \cup \boldsymbol{S_s}$
return $\boldsymbol{S}$

---

**Algorithm 4** Query Aware Index Search

**Input:** $\boldsymbol{Q}, \boldsymbol{K}, \gamma$

*# Compute estimated attention scores using pooled $\boldsymbol{Q}$ and $\boldsymbol{K}$*
$\bar{\boldsymbol{Q}} \leftarrow \text{pool}(\boldsymbol{Q}),\ \bar{\boldsymbol{K}} \leftarrow \text{pool}(\boldsymbol{K})$
$\bar{\boldsymbol{A}} \leftarrow \text{softmax}\left(\bar{\boldsymbol{Q}}\bar{\boldsymbol{K}}^\top/\sqrt{d}\right)$

*# Flatten and normalize attention map*
$\bar{\boldsymbol{A}} \leftarrow \text{flatten}(\bar{\boldsymbol{A}}/\sum_{i,j} \bar{\boldsymbol{A}}[i,j])$

*# Sort attention scores*
$\boldsymbol{I_a} \leftarrow \text{argsort}(\bar{\boldsymbol{A}})$

*# Obtain the minimum computational budget making the sum of the scores exceeds $\gamma$*
$K \leftarrow \min\{k : \sum_{i \in \boldsymbol{I_a}[1:k]} \bar{\boldsymbol{A}}[i] \geq \gamma\}$

*# Get final index set*
$\boldsymbol{S} \leftarrow \boldsymbol{I_a}[1:K]$
return $\boldsymbol{S}$

---

**Models** We utilize four state-of-the-art LLMs known for their proficiency in handling long-context tasks: (i) Meta-Llama-3.1-8B-Instruct-128k (Dubey et al., 2024) (**LLaMA**) (ii) GLM-4-9B-Chat-1024k (Zeng et al., 2024) (**GLM**) (iii) Yi-9B-200K (Young et al., 2024) (**Yi**) (iv) Qwen2-7B-Instruct-128k (Yang et al., 2024) (**Qwen**), where Yi is a pre-trained model and all others are instruct models. We use the default chat template for all instruct models in the subsequent experiments.

**Datasets** We evaluate the models on two datasets, each offering unique challenges in long-context understanding: (i) **RULER** (Hsieh et al., 2024): a synthetic benchmark dataset created to evaluate

long-context LLMs with customizable sequence lengths and task complexities. It extends the basic needle-in-a-haystack test as well as introduces new task categories such as multi-hop tracing and aggregation. (ii) **Infinite Bench** (Zhang et al., 2024c): a benchmark dataset designed to test LLMs' understanding of long dependencies within extensive contexts, with an average token count of 214k. It includes 10 synthetic and real-world tasks across various domains.

**Implementation Details**  Our experiments are conducted in a computing environment equipped with a single NVIDIA A100 GPU with 80GB of memory. For our experiments, we implement a custom pipeline using PyTorch, building on *FlashAttention* (Dao, 2024), to ensure efficient attention mechanisms over long-context inputs. Our implementation leverages *Triton* (Tillet et al., 2019) for optimizing the performance of GPU-accelerated computations and uses a *block_size* = 128 in all experiments. We choose the last *block_size* query vector as the representative query set $\hat{Q}$ for both sparse pattern determination and *Vertical-Slash* sparse index selection, which allows the representative attention score to be computed only once and reused in both attention pattern determination and sparse index selection. The sparse pattern threshold $\tau$ is set to 0.1 for all models. Additionally, by adjusting the parameter $\gamma$, we assign a tailored computational budget to each attention head in real-time for each input scenario. We set $\gamma = 0.9$ for Yi-9B-200k and Qwen2-7B-Instruct model, and 0.95 for the other models. To ensure that all attention heads work normally, we retain the first and last key blocks of each query block, while also requiring each attention head to compute at least 1024 tokens. All experiments were conducted using greedy decoding to maintain consistency across results.

**Baselines**  We evaluate our method against three strong baseline approaches to highlight its efficiency and performance in long-context tasks: 1) **FlashAttention** (Dao, 2024): A highly optimized attention mechanism that exploits underlying hardware optimizations and efficient memory access patterns, which significantly reduces the computational and memory overhead of attention operations. 2) **StreamingLLM** (Xiao et al., 2024b): A sparse attention mechanism that combines global attention sink tokens with dilated local attention windows. We use a dilated window configuration with an interval of 1, encompassing 1,000 global tokens and 8,000 local windows. 3) **MInference** (Jiang et al., 2024a): An efficient sparse attention method for handling long sequences, employing three distinct sparse attention patterns. According to the recommended settings from the original paper, we utilize offline search attention patterns and dynamically generated indexes to compute each attention head separately. Additionally, we include more baseline comparisons for a comprehensive evaluation in Appendix D.

All baselines use sparse computation during pre-filling, switching to dense computation during the decoding stage. This ensures a fair comparison across different methods and highlights the balance between latency and performance.

## 4.2 MAIN RESULT

This section presents a comparative analysis of various attention mechanisms. The evaluation includes Full Attention, Miniference, Streaming LLM, and our proposed method. The comparison aims to demonstrate our approach's efficiency and effectiveness against existing methods.

**RULER**  To demonstrate the potential of our approach for long-context large language models, we conduct evaluations using the RULER benchmark. Table 1 illustrates that Streaming LLM's performance significantly deteriorates as context length increases, while MInference shows suboptimal performance on certain models. In contrast, our method consistently preserves various models' performance across multiple context lengths. Notably, our approach can enhance models' capabilities in some scenarios while accelerating the computational process. Detailed latency comparisons are provided in Appendix C.

**Infinite Bench**  To further demonstrate our approach's potential for long-context LLMs, we conduct evaluations using the state-of-the-art Infinite Bench challenge. Table 2 shows that our method preserves most of the model's performance in retrieval and QA tasks, while maintaining efficacy in complex mathematical and coding tasks.

Table 1: Performance comparison of different methods on various models and sequence lengths on RULER. The best results are highlighted in bold.

| Models | Methods | 4k | 8k | 16k | 32k | 64k | 128k | Avg |
|---|---|---|---|---|---|---|---|---|
| LLaMA | Full-attn | 95.67 | 93.75 | 93.03 | 87.26 | 84.37 | 78.13 | 88.70 |
| | Streaming LLM | 95.43 | **93.99** | 74.76 | 48.56 | 26.20 | 30.77 | 61.62 |
| | MInference | **95.67** | **93.99** | 93.27 | 86.54 | **84.86** | 58.17 | 85.42 |
| | Ours | 95.43 | 93.51 | **94.71** | **89.42** | 82.93 | **79.09** | **89.18** |
| GLM | Full-attn | 93.75 | 93.03 | 89.66 | 90.63 | 85.34 | 81.97 | 89.06 |
| | Streaming LLM | **93.75** | 92.79 | 76.92 | 56.97 | 40.14 | 34.86 | 65.91 |
| | MInference | **93.75** | **93.03** | **90.38** | 89.90 | 85.10 | 82.93 | 89.18 |
| | Ours | 93.51 | 91.83 | 89.90 | **91.35** | **86.06** | **83.41** | **89.34** |
| Yi | Full-attn | 92.79 | 86.06 | 85.34 | 76.93 | 69.47 | 66.35 | 79.49 |
| | Streaming LLM | 93.03 | 83.41 | 65.38 | 51.68 | 39.18 | 34.61 | 61.22 |
| | MInference | 92.79 | 85.58 | 82.69 | **73.32** | 63.94 | **57.69** | 76.00 |
| | Ours | **93.27** | **86.78** | 83.89 | 72.36 | **64.66** | 56.97 | **76.32** |
| Qwen | Full-attn | 89.90 | 88.70 | 80.77 | 79.33 | 56.49 | 17.79 | 68.83 |
| | Streaming LLM | 90.14 | 88.94 | 57.93 | 43.03 | 25.48 | 12.26 | 52.96 |
| | MInference | 89.90 | 88.70 | 79.33 | 78.61 | 49.04 | 10.82 | 66.07 |
| | Ours | **90.39** | **89.91** | **83.17** | **81.25** | **59.14** | **20.67** | **70.75** |

Table 2: Performance comparison of different methods on various models and tasks on Infinite Bench. The **bold**/underlined numbers indicate the first/second highest value in each column.

| Models | Methods | En.Sum | En.QA | En.MC | En.Dia | Zh.QA | Code.Debug | Math.Find | Retr.PassKey | Retr.Number | Retr.KV | Avg |
|---|---|---|---|---|---|---|---|---|---|---|---|---|
| LLaMA | Full-attn | 31.91 | 25.92 | 69.43 | 21.50 | 31.95 | 16.75 | 24.29 | 99.15 | 99.66 | 60.00 | 48.06 |
| | Streaming LLM | 30.15 | 10.15 | 41.05 | 8.50 | 22.38 | 8.63 | 17.71 | 2.71 | 5.93 | 0.00 | 14.72 |
| | Minference | 31.04 | 22.00 | 63.76 | 14.50 | 28.70 | 5.33 | 27.43 | 56.78 | 77.12 | 14.00 | 34.06 |
| | Ours | **31.82** | **24.82** | **69.43** | **19.50** | **35.46** | **16.75** | **31.14** | **98.64** | **99.83** | **44.00** | **47.14** |
| GLM | Full-attn | 28.60 | 20.70 | 42.36 | 35.00 | 15.69 | 32.99 | 26.29 | 99.15 | 100.00 | 19.00 | 41.98 |
| | Streaming LLM | 28.15 | 11.58 | 31.00 | 18.50 | 13.77 | 26.65 | 20.29 | 15.08 | 100.00 | 6.60 | 27.16 |
| | Minference | 27.27 | 19.42 | **44.98** | 29.50 | 15.83 | **36.29** | 23.71 | **100.00** | 100.00 | 48.60 | **44.56** |
| | Ours | **28.90** | **20.27** | 42.36 | 29.50 | 15.55 | 33.50 | 23.71 | 99.15 | 100.00 | **55.40** | 44.83 |
| Yi | Full-attn | 8.32 | 11.47 | 65.50 | 1.50 | 17.49 | 20.81 | 23.14 | 100.00 | 99.66 | 29.00 | 37.69 |
| | Streaming LLM | 6.01 | 11.21 | 42.79 | **3.50** | 17.15 | 19.54 | 22.86 | 10.17 | 36.10 | 3.60 | 17.29 |
| | Minference | 6.06 | 10.27 | 62.45 | 2.00 | **17.74** | **24.37** | **24.86** | **100.00** | **97.63** | **29.00** | **37.44** |
| | Ours | **6.62** | **11.89** | **65.07** | 2.50 | 16.87 | 19.54 | 24.00 | 99.83 | 91.53 | 24.00 | 36.18 |
| Qwen | Full-attn | 4.65 | 5.57 | 34.50 | 9.00 | 11.27 | 24.87 | 24.57 | 95.42 | 75.42 | 0.00 | 29.53 |
| | Streaming LLM | **18.54** | 6.43 | **39.30** | 12.50 | 10.94 | **24.11** | 28.00 | 30.85 | 61.69 | 0.00 | 23.24 |
| | Minference | 7.45 | 3.94 | 14.85 | 4.50 | 10.17 | 13.20 | 30.00 | 83.90 | 58.47 | 0.00 | 22.65 |
| | Ours | 14.27 | **6.55** | 34.06 | **13.50** | **11.51** | 20.81 | **30.86** | **96.95** | **72.20** | 0.00 | **30.07** |

**Performance vs. Latency** Our method leverages online search characteristics, enabling seamless application to various models and flexible adjustments between computational speed and performance. Tuning the $\gamma$ parameter balances model efficacy and output latency: decreasing $\gamma$ accelerates processing while increasing $\gamma$ preserves model quality. Figure 4 compares our approach with MInference, demonstrating that our method achieves better performance with lower latency. A more detailed latency analysis can be found in Appendix G.

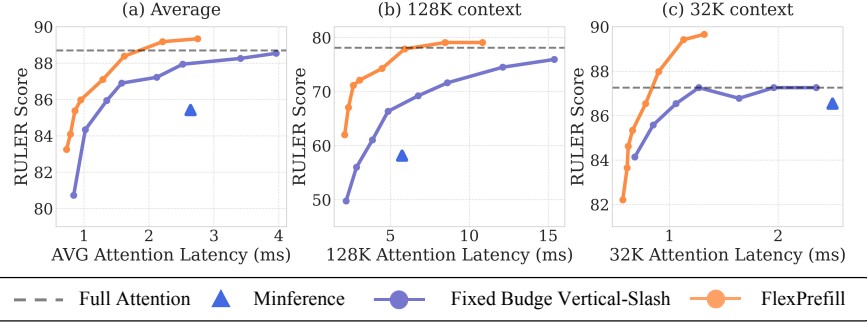

Figure 4: Comparison of our method with MInference and Fixed budget Vertical-Slash attention, showing the trade-off between model performance and attention latency. Our method consistently outperforms MInference and Fixed Budget Vertical-Slash across different attention latencies. More details are provided in Appendix E

### 4.3 ABLATIONS

We conduct various ablation studies to analyze the components of our proposed method and understand the impact of each configuration, with some details provided in the Appendix due to space constraints. We begin by validating the necessity of dynamically computing budget allocations. We explore the significance of the *Query-Aware* attention head and the selection of threshold $\tau$. We also investigate the impact of various minimum computational budget allocations on the robustness of the sparse attention mechanism.

Furthermore, we find that the block size of sparse attention does not significantly impact model effectiveness (Appendix F.2). We observe that different representative query set $\hat{Q}$ minimally affect the *Query-Aware* pattern determination but substantially influence the index selection of the *Vertical-Slash* head (Appendix F.3). This observation justifies the use of the last $block\_size$ query vectors as $\hat{Q}$. We evaluate an alternative index construction implementation for Algorithm 4 and observe that it has negligible impact on model performance (Appendix F.4). Finally, we study the effect of limiting the maximum computational budget of the attention heads and discover that the performance of some models saturates with increased computation, indicating the potential for further acceleration (Appendix F.6).

**Fixed Budget vs. Dynamic Budget**  We experiment with fixed and dynamic budget allocations. Fixed budget uses the same computational budget for all attention heads across all layers. Results in Figure 4 show that dynamic settings lead to improved performance and a better balance between inference speed and model effectiveness compared to static budget allocation.

**Query-Aware head threshold**  We evaluate model performance with and without *Query-Aware* attention pattern and examine different $\tau$ impacts on performance. Figure 5 demonstrates that enabling *Query-Aware* attention head at an appropriate $\tau$ strengthens the model without increasing computation. However, if $\tau$ is set too large, some attention heads with inaccurate attention estimation are recognized as *Query-Aware* heads, potentially degrading model performance. More detailed results can be found in Appendix F.1.

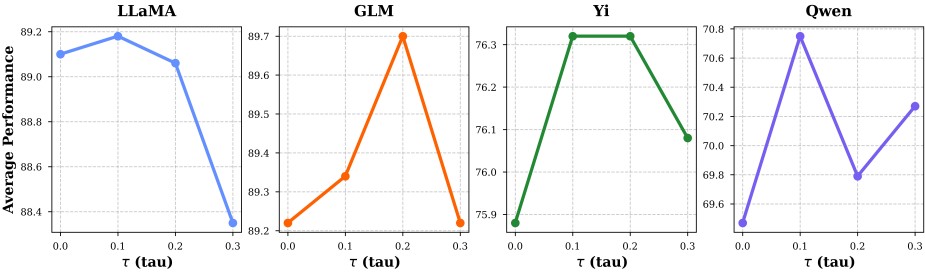

Figure 5: Comparison of our method under different thresholds $\tau$ indicates that an appropriate threshold $\tau$ improves model performance.

**Minimum Budget Limit**  We perform ablation experiments to analyze the minimum computational budgets required for each attention head. Our findings reveal that setting a minimum budget threshold enhances performance and prevents the collapse of attention heads with extremely high sparsity ratios. More detailed results can be found in Appendix F.5 and Table 10.

To further analyze the components of our approach, we visualize the distribution of sparse patterns between different layers (Appendix H), the sparse masks of *Query-Aware* and *Vertical-Slash* heads in our approach (Appendix I), and the distribution of sparsity ratios for different layers and heads (Appendix J), respectively. This visualizes the flexibility of our approach and the significance of each component.

## 5 RELATED WORKS

**Long Context LLMs** Ongoing advancements in engineering and algorithmic design continue to push the boundaries of long contexts large language models (LLMs), enabling them to tackle increasingly complex tasks. Some methods extend the context length of the mode by gathering extensive long-text datasets and persistently pre-training or fine-tuning the models (Fu et al., 2024b; Chen et al., 2024; Xiong et al., 2024). Given that many modern LLMs employ RoPE-based positional embeddings (Su et al., 2021), various innovative positional embedding techniques have been introduced to extend model lengths, exemplified by Peng et al. (2024), Ding et al. (2024), and Zhang et al. (2024d). Moreover, approaches leveraging external memory or retrieval augmentation have been proposed to enhance long context processing capabilities (Mohtashami & Jaggi, 2023; Tworkowski et al., 2023; Xu et al., 2024).

**LLM Inference Acceleration** Given the quadratic time complexity of self-attention relative to sequence length, accelerating inference for long contexts is crucial for LLMs. Some strategies optimize the original attention computation through algorithms synergized with hardware features, including FlashAttention (Dao et al., 2022; Dao, 2024; Shah et al., 2024) and RingAttention (Liu et al., 2023; Brandon et al., 2023). Additional methods facilitate attention acceleration by reducing context length, such as retrieval-based techniques (Mohtashami & Jaggi, 2023; Wang et al., 2023; Xu et al., 2024; Munkhdalai et al., 2024) and compression-based approaches (Li et al., 2023; Jiang et al., 2024b; Pan et al., 2024). Strategies addressing the quadratic complexity of attention computations include employing recurrent mechanisms to aggregate information (Zhang et al., 2024b; Bulatov et al., 2023; Martins et al., 2022), using State Space Models (Gu et al., 2022; Gu & Dao, 2023; Lieber et al., 2024), and exploring innovative model architectures (Sun et al., 2023; Peng et al., 2023; Beck et al., 2024). Given the inherent sparsity in attention mechanisms, numerous sparse attention methods have been proposed. Many focus on fixed attention patterns, such as shifted sparse attention (Chen et al., 2024), sink attention (Xiao et al., 2024b), and other approaches (Child et al., 2019; Beltagy et al., 2020; Zaheer et al., 2020; Shi et al., 2021; Ding et al., 2023). Modern LLMs like mistral (Jiang et al., 2023) and phi3 (Abdin et al., 2024) also employ fixed sparse attention patterns, including sliding window attention. Han et al. (2024) and Xiao et al. (2024a) further extend the sink attention approach. Additionally, other studies, such as those by Likhosherstov et al. (2021), Liu et al. (2021), and Jiang et al. (2024a), highlight the presence of dynamic sparse patterns in models.

For the decoding phase, KV caching techniques that store past keys and values are prevalent, introducing unique challenges for LLM inference acceleration. Some methods optimize computational processes or KV caches management more efficiently, such as FlashDecoding (Dao et al., 2023; Hong et al., 2023) and PagedAttention (Kwon et al., 2023). Other approaches aim to reduce KV cache sizes through quantization (Liu et al., 2024a; Kang et al., 2024; Liu et al., 2024b), token merging (Nawrot et al., 2024; Li et al., 2024), or KV discarding (Zhang et al., 2023; Dong et al., 2024). Furthermore, sparse attention methods are widely applied in the decoding stage. Some techniques retain the entire KV cache but utilize only a subset for sparse attention computations Ribar et al. (2024); Tang et al. (2024); Zhang et al. (2024a), while others diminish the KV cache via specific sparse attention patterns Ge et al. (2024); Xiao et al. (2024b); Zhang et al. (2023).

## 6 CONCLUSION

In this paper, we introduced FlexPrefill, a novel flexible sparse attention mechanism designed to adapt in real-time for efficient long-sequence pre-filling in LLMs. Our approach, comprising Query-Aware Sparse Pattern Determination and Cumulative-Attention Based Index Selection, addresses the limitations of previous methods by dynamically optimizing sparse patterns and ratios for each attention head based on inputs. Extensive experiments across state-of-the-art LLMs and challenging long-context benchmarks demonstrate that FlexPrefill consistently preserves or enhances model performance while significantly improving computational efficiency. The adaptive nature of our method allows for a better balance between speed and accuracy, outperforming prior methods across various scenarios. As LLMs continue to evolve and tackle increasingly complex long-context tasks, FlexPrefill offers a promising solution for maintaining high performance while managing computational resources effectively. Future work could explore further optimizations and applications of this adaptive approach in diverse model architectures.

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

# A   RATIONALE

In practice, we can achieve the multi-objective optimization goal (Equation (1)) by setting a tolerance rate $\gamma_S$ for the size of the selected subset $|\boldsymbol{S}|$. Given this constraint, the goal is to minimize the attention difference $|\boldsymbol{A}(\boldsymbol{Q}, \boldsymbol{K}, \boldsymbol{V}) - \boldsymbol{A}(\boldsymbol{Q}, \boldsymbol{K}, \boldsymbol{V}, \boldsymbol{S})|$ to maintain the effectiveness of the sparse attention mechanism while reducing its computational cost. This can be formulated as the following constrained optimization problem:

$$
\begin{aligned}
\min_{\boldsymbol{S}} \quad & |\boldsymbol{A}(\boldsymbol{Q}, \boldsymbol{K}, \boldsymbol{V}) - \boldsymbol{A}(\boldsymbol{Q}, \boldsymbol{K}, \boldsymbol{V}, \boldsymbol{S})| \\
\text{subject to} \quad & |\boldsymbol{S}| \leq \gamma_S \\
& \boldsymbol{S} \subseteq \{(i, j) \,|\, 1 \leq j \leq i, \, i, j \in \mathbb{Z}\}
\end{aligned}
\tag{4}
$$

We now provide a deeper understanding of how optimizing the sparse attention objective (Equation 3) leads to achieving the goal (Equation 4). For brevity, let us focus on the optimization problem for a single query position, omit the subscript $i$, and consider the $i^{\text{th}}$ query position in the following analysis. The standard attention computation for this query position is given by: $\boldsymbol{A}(\boldsymbol{Q}, \boldsymbol{K}, \boldsymbol{V}) = \sum_{j=1}^{L} \frac{e^{x_j} v_j}{\sum_{j'=1}^{L} e^{x_{j'}}}$ and the sparse attention computation for this subset is: $\boldsymbol{A}(\boldsymbol{Q}, \boldsymbol{K}, \boldsymbol{V}, \boldsymbol{S}) = \sum_{j \in S} \frac{e^{x_j} v_j}{\sum_{j' \in S} e^{x_{j'}}}$. Let the attention scores be defined as $a_j = \frac{e^{x_j}}{\sum_{j'=1}^{L} e^{x_{j'}}}$. The difference between the full attention and sparse attention computations can be expressed as:

$$
\begin{aligned}
\boldsymbol{A}(\boldsymbol{Q}, \boldsymbol{K}, \boldsymbol{V}) - \boldsymbol{A}(\boldsymbol{Q}, \boldsymbol{K}, \boldsymbol{V}, \boldsymbol{S}) &= \sum_{j=1}^{L} \frac{e^{x_j} v_j}{\sum_{j'=1}^{L} e^{x_{j'}}} - \sum_{j \in S} \frac{e^{x_j} v_j}{\sum_{j' \in S} e^{x_{j'}}} \\
&= \sum_{j=1}^{L} a_j v_j - \frac{1}{\sum_{j \in S} a_j} \sum_{j' \in S} a_{j'} v_{j'} \\
&= \sum_{j \notin S} a_j v_j - \left( \frac{1}{\sum_{j \in S} a_j} - 1 \right) \sum_{j' \in S} a_{j'} v_{j'}
\end{aligned}
\tag{5}
$$

Let $a_S = \sum_{j \in S} a_j$. Then, $a_k \leq a_S$ for $k \in S$, and $a_k \leq 1 - a_S$ for $k \notin S$. Substituting these bounds into Equation 5, we obtain an upper bound for the absolute difference between full and sparse attention:

$$
\begin{aligned}
\boldsymbol{A}(\boldsymbol{Q}, \boldsymbol{K}, \boldsymbol{V}) - \boldsymbol{A}(\boldsymbol{Q}, \boldsymbol{K}, \boldsymbol{V}, \boldsymbol{S}) &\leq \sum_{j \notin S} a_j |v_j| + \left( \frac{1}{a_S} - 1 \right) \sum_{j \in S} a_j |v_j| \\
&\leq (1 - a_S) \sum_{j \notin S} |v_j| + \left( \frac{1}{a_S} - 1 \right) a_S \sum_{j \in S} |v_j| \\
&= (1 - a_S) \sum_{j=1}^{L} |v_j|
\end{aligned}
\tag{6}
$$

To minimize the upper bound of the error, we can maximize the sum of normalized attention scores $a_S$ in the selected subset, subject to a constraint on the subset size:

$$
\max_{S \subseteq [L]} \sum_{j \in S} \frac{\exp(x_j)}{\sum_{j'=1}^{L} \exp(x'_j)}, \quad \text{subject to} \quad |S| \leq \gamma_S
\tag{7}
$$

By applying duality principles (Appendix B for a detailed derivation), we can transform the primal objective (Equation (7)) into its dual form, which minimizes the subset size subject to a constraint on the sum of normalized attention scores. Given a tolerance $0 < \gamma_a < 1$, the dual optimization

objective for a single query position becomes:

$$\min_{S \subseteq [L]} |S|, \quad \text{subject to} \quad \sum_{j \in S} \frac{\exp(x_j)}{\sum_{j'=1}^{L} \exp(x_{j'})} \geq \gamma_a \tag{8}$$

The general objective for all query positions, as shown in Equation (3), can be obtained by aggregating the objectives for individual query positions (Equation (8)) across all $i \in [n]$

## B  DUAL FORM: BALANCING SUBSET SIZE AND ATTENTION SCORES

To better understand the relationship between the original objective and its dual form, we derive the dual objective step by step. For brevity, let us focus on the optimization problem for a single query position, omit the subscript $i$, and consider the $i^{\text{th}}$ query position in the following analysis.

**Step 1: Binary Variables**   If $z_j = 1$ if $j \in S$ and $z_j = 0$ otherwise, the primal objective is:

$$\max_{z_j \in \{0,1\}} \sum_{j=1}^{L} z_j \frac{\exp(x_j)}{\sum_{j'=1}^{L} \exp(x_{j'})}, \quad \text{subject to} \quad \sum_{j=1}^{L} z_j \leq \gamma_S$$

**Step 2: Relax Binary Variables**   Relaxing to continuous variables:

$$\max_{0 \leq z_j \leq 1} \sum_{j=1}^{L} z_j \frac{\exp(x_j)}{\sum_{j'=1}^{L} \exp(x_{j'})}, \quad \text{subject to} \quad \sum_{j=1}^{L} z_j \leq \gamma_S$$

**Step 3: Lagrangian Formulation**   Let's form the Lagrangian for the primal problem:

$$L(z, \lambda, \mu, \nu) = -\sum_{j=1}^{L} z_j \frac{\exp(x_j)}{\sum_{j'=1}^{L} \exp(x_{j'})} + \lambda(\sum_{j=1}^{L} z_j - \gamma_S) - \sum_{j=1}^{L} \mu_j z_j + \sum_{j=1}^{L} \nu_j (z_j - 1)$$

$\lambda \geq 0$ is the multiplier for size constraint $\mu_j \geq 0$ are multipliers for $z_j \geq 0$ and $\nu_j \geq 0$ are multipliers for $z_j \leq 1$.

**Step 4: KKT Conditions**   The KKT conditions give:

1. Stationarity:
$$\frac{\partial L}{\partial z_j} = -\frac{\exp(x_j)}{\sum_{j'=1}^{L} \exp(x_{j'})} + \lambda - \mu_j + \nu_j = 0$$

2. Complementary slackness:
$$\lambda(\sum_{j=1}^{L} z_j - \gamma_S) = 0$$
$$\mu_j z_j = 0$$
$$\nu_j (z_j - 1) = 0$$

**Step 5: Optimal Solution Structure**   From stationarity condition:

$$z_j = \begin{cases} 1 & \text{if } \frac{\exp(x_j)}{\sum_{j'=1}^{L} \exp(x_{j'})} > \lambda \\ 0 & \text{if } \frac{\exp(x_j)}{\sum_{j'=1}^{L} \exp(x_{j'})} < \lambda \end{cases}$$

**Step 6: Duality**   The dual problem becomes finding minimum $|S|$ (or equivalently $\sum_{j=1}^{L} z_j$) that achieves required attention mass $\gamma_a$. This gives us:

$$\min_{0 \leq z_j \leq 1} \sum_{j=1}^{L} z_j \quad \text{subject to} \sum_{j=1}^{L} z_j \frac{\exp(x_j)}{\sum_{j'=1}^{L} \exp(x_{j'})} \geq \gamma_a$$

**Understanding the Duality Transformation** The transformation from primal to dual form arises naturally from the optimality conditions of the Lagrangian. Given the primal problem $\max_{z_j} \sum_{j=1}^{L} z_j \frac{\exp(x_j)}{\sum_{j'=1}^{L} \exp(x_{j'})}$ subject to $\sum_{j=1}^{L} z_j \leq \gamma_S$, the Lagrange multiplier $\lambda$ associated with the size constraint effectively becomes a threshold on attention scores. At optimality, positions are selected (i.e., $z_j = 1$) if and only if their normalized attention score $\frac{\exp(x_j)}{\sum_{j'=1}^{L} \exp(x_{j'})}$ exceeds $\lambda$. This naturally leads to the dual formulation $\min \sum_{j=1}^{L} z_j$ subject to $\sum_{j=1}^{L} z_j \frac{\exp(x_j)}{\sum_{j'=1}^{L} \exp(x_{j'})} \geq \gamma_a$, where $\gamma_a$ corresponds to the optimal attention mass achieved in the primal problem. The equivalence holds by strong duality, as the primal problem's convex relaxation ensures zero duality gap. The dual objective minimizes the size of the selected subset $S$ while ensuring that the sum of normalized attention scores in the subset is greater than or equal to a threshold $\gamma_a$. This is precisely the same objective defined in Equation (3),

By deriving the dual form and showing that it is equivalent to our main objective, we establish a strong connection between the Equation (7) and Equation (3) formulations. This connection highlights the inherent trade-off between the subset size and the attention scores, and it provides a solid theoretical foundation for our optimization goal.

## C  DETAILED LATENCY OF DIFFERENT METHODS

To highlight the efficiency and performance advantages of our method, we present detailed attention computation latency comparisons on the RULER dataset. Specifically, we evaluate the average latency of a single attention function call for long sequences (64k and 128k tokens) and the overall average latency across all sequence lengths. As shown in Table 3, our FlexPrefill method outperforms competing approaches by achieving better performance while maintaining lower latency.

Table 3: Latency comparison of different methods on various models and sequence lengths on RULER.

| Models | Methods | 64k Time (ms) | 128k Time (ms) | Avg Time (ms) | Avg Score |
|---|---|---|---|---|---|
| LLaMA | Full-attn | 157.88 | 658.83 | 144.97 | 88.70 |
|  | Streaming LLM | 88.41 | 180.40 | 56.08 | 61.62 |
|  | MInference | 152.56 | 215.53 | 116.63 | 85.42 |
|  | Ours ($\gamma = 0.9$) | 61.92 | 185.75 | 50.66 | 88.38 |
|  | Ours ($\gamma = 0.95$) | 85.4 | 271.07 | 70.61 | 89.18 |
| GLM | Full-attn | 267.26 | 1066.21 | 237.65 | 89.06 |
|  | Streaming LLM | 88.1 | 179.95 | 55.72 | 65.91 |
|  | MInference | 295.95 | 493.12 | 226.71 | 89.18 |
|  | Ours ($\gamma = 0.9$) | 87.57 | 279.82 | 71.89 | 88.58 |
|  | Ours ($\gamma = 0.95$) | 120.07 | 408.74 | 100.75 | 89.34 |
| Yi | Full-attn | 243.17 | 1018.07 | 223.53 | 79.49 |
|  | Streaming LLM | 107.45 | 218.96 | 68.40 | 61.22 |
|  | MInference | 171.48 | 305.69 | 138.86 | 76.00 |
|  | Ours ($\gamma = 0.85$) | 81.77 | 244.90 | 53.85 | 74.28 |
|  | Ours ($\gamma = 0.9$) | 102.32 | 319.43 | 83.30 | 76.32 |
| Qwen | Full-attn | 137.53 | 569.86 | 125.67 | 68.83 |
|  | Streaming LLM | 77.62 | 157.51 | 49.46 | 52.96 |
|  | MInference | 220.51 | 306.47 | 162.25 | 66.07 |
|  | Ours ($\gamma = 0.85$) | 49.10 | 133.19 | 38.00 | 70.39 |
|  | Ours ($\gamma = 0.9$) | 61.51 | 173 | 47.62 | 70.75 |

## D  COMPARISON WITH ADDITIONAL BASELINES

To further assess the efficiency of our proposed method, we provide performance comparisons against additional baselines. Specifically, we evaluated LM-Infinite (Han et al., 2024), InfLLM (Xiao et al., 2024a), and MoA (Fu et al., 2024a) using the Llama-3-8B-Instruct-262k (Pekelis et al., 2024) models. Additionally, we assessed HIP (Lee et al., 2024) using the Meta-Llama-3.1-8B-Instruct-128k (Dubey et al., 2024) model. All evaluations were performed on the RULER dataset.

As shown in Table 4, our method consistently outperforms these baselines in both performance and inference speed.

Table 4: Performance comparison of additional methods on various models and sequence lengths on RULER

| Models | Methods | 4k | 8k | 16k | 32k | 64k | 128k | Avg | 128k speedup |
|---|---|---|---|---|---|---|---|---|---|
| | Full-attn | 91.11 | 89.18 | 88.94 | 84.37 | 81.01 | 77.88 | 85.42 | - |
| | MInference | 90.87 | 89.90 | 90.38 | 82.21 | 81.73 | 78.13 | 85.54 | 1.89x |
| | MoA | 91.35 | 88.22 | 85.58 | 83.9 | 81.97 | 76.44 | 84.58 | 1.31x |
| LLaMA-3 | InfLLM | 89.4 | 79.8 | 70.1 | 55.6 | 43 | 39.5 | 62.9 | 1.51x |
| | LM-Infinite | 91.11 | 54.81 | 38.94 | 21.39 | 14.66 | OOM | 35.82 | - |
| | Ours ($\gamma = 0.9$) | 89.18 | 89.18 | 89.66 | 83.65 | 79.33 | 77.64 | 84.78 | 3.47x |
| | Ours ($\gamma = 0.95$) | 90.87 | 89.9 | 90.38 | 82.21 | 81.25 | 79.81 | 85.74 | 2.43x |
| | Full-attn | 95.67 | 93.75 | 93.03 | 87.26 | 84.37 | 78.13 | 88.70 | - |
| | MInference | 95.67 | 93.99 | 93.27 | 86.54 | 84.86 | 58.17 | 79.09 | 3.05x |
| LLaMA-3.1 | HiP | 96.0 | 94.7 | 94.1 | 89.7 | 79.9 | 58.2 | 85.4 | 1.70x |
| | Ours ($\gamma = 0.9$) | 95.43 | 93.27 | 94.23 | 87.98 | 81.49 | 77.89 | 88.38 | 3.49x |
| | Ours ($\gamma = 0.95$) | 95.43 | 93.51 | 94.71 | 89.42 | 82.93 | 79.09 | 89.18 | 2.43x |

# E  PERFORMANCE-LATENCY TRADE-OFF WITH DIFFERENT $\gamma$

In Figure 4, we plot model performance against the average prefill time for a single attention head. The average latency of the full attention model is reported as follows: 4.5 ms (averaged across all input lengths), 20.58 ms (for an input length of 128k), and 1.19 ms (for an input length of 32k). Our proposed method demonstrates varying latency depending on the parameter $\gamma$, while maintaining comparable performance. We provide a comprehensive comparison for different values of $\gamma$, facilitating an empirical selection process. As shown in Table 5, reducing $\gamma$ results in faster speeds but with a slight trade-off in performance. In practice, $\gamma$ can be flexibly adjusted to meet specific performance or speed requirements.

Table 5: Performance comparison of different $\gamma$ for Llama-3.1-8B model

| $\gamma$ | 4k | 8k | 16k | 32k | 64k | 128k | Avg | 128k speedup |
|---|---|---|---|---|---|---|---|---|
| 1 (Full-Attn) | 95.67 | 93.75 | 93.03 | 87.26 | 84.37 | 78.13 | 88.70 | - |
| 0.97 | 95.43 | 93.99 | 94.47 | 89.66 | 83.41 | 79.09 | 89.34 | 1.89x |
| 0.95 | 95.43 | 93.51 | 94.71 | 89.42 | 82.93 | 79.09 | 89.18 | 2.43x |
| 0.9 | 95.43 | 93.27 | 94.23 | 87.98 | 81.49 | 77.89 | 88.38 | 3.49x |
| 0.85 | 95.91 | 93.51 | 92.31 | 86.54 | 80.05 | 74.28 | 87.10 | 4.60x |
| 0.8 | 96.39 | 92.79 | 92.31 | 87.74 | 80.05 | 71.39 | 86.78 | 5.20x |
| 0.75 | 95.19 | 93.51 | 92.31 | 86.06 | 78.13 | 70.43 | 85.94 | 6.75x |
| 0.7 | 95.67 | 93.03 | 91.35 | 84.62 | 76.44 | 71.15 | 85.38 | 7.75 |
| 0.65 | 95.91 | 92.55 | 91.35 | 83.65 | 74.04 | 67.07 | 84.09 | 8.79x |
| 0.6 | 95.67 | 93.27 | 91.83 | 82.21 | 74.52 | 62.02 | 83.25 | 9.81x |

# F  ADDITIONAL ABLATION STUDY

## F.1  QUERY-AWARE HEAD THRESHOLD

We evaluate the impact of different $\tau$ on the model performance and provide the detailed performance of the model under different context lengths in Table 6.

## F.2  TRITON BLOCK SIZE

We explore the impact of adjusting Triton block sizes, specifically comparing block sizes of 64 and 128. The results in Table 7 show that the block size does not have a significant effect on the performance of the model, so different block sizes can be flexibly selected according to different hardware.

Table 6: Performance comparison of different models using different *Query-Aware* head threshold $\tau$ on the RULER dataset.

| model | $\tau$ | 4k | 8k | 16k | 32k | 64k | 128k | avg |
|-------|--------|------|------|------|------|------|------|------|
| LLaMA | 0 | 95.43 | 93.75 | 94.71 | 89.67 | 82.45 | 78.61 | 89.10 |
|       | 0.1 | 95.43 | 93.51 | 94.71 | 89.42 | 82.93 | 79.09 | 89.18 |
|       | 0.2 | 95.19 | 93.75 | 94.95 | 89.42 | 83.17 | 77.88 | 89.06 |
|       | 0.3 | 95.19 | 93.27 | 93.99 | 89.66 | 82.21 | 75.96 | 88.38 |
| GLM   | 0 | 93.75 | 91.35 | 90.14 | 91.83 | 86.78 | 81.49 | 89.22 |
|       | 0.1 | 93.51 | 91.83 | 89.90 | 91.35 | 86.06 | 83.41 | 89.34 |
|       | 0.2 | 93.99 | 92.31 | 90.38 | 91.11 | 86.78 | 83.66 | 89.70 |
|       | 0.3 | 93.51 | 92.07 | 89.66 | 90.14 | 87.26 | 82.69 | 89.22 |
| Yi    | 0 | 93.51 | 87.26 | 83.41 | 71.88 | 62.74 | 56.49 | 75.88 |
|       | 0.1 | 93.27 | 86.78 | 83.89 | 72.36 | 64.66 | 56.97 | 76.32 |
|       | 0.2 | 93.75 | 86.54 | 83.41 | 73.80 | 62.50 | 57.93 | 76.32 |
|       | 0.3 | 93.27 | 85.82 | 84.37 | 73.56 | 62.02 | 57.45 | 76.08 |
| Qwen  | 0 | 89.42 | 87.50 | 82.21 | 81.73 | 56.97 | 18.99 | 69.47 |
|       | 0.1 | 90.39 | 89.91 | 83.17 | 81.25 | 59.14 | 20.67 | 70.75 |
|       | 0.2 | 90.63 | 88.22 | 82.21 | 80.29 | 57.93 | 19.47 | 69.79 |
|       | 0.3 | 91.83 | 88.46 | 83.41 | 80.05 | 58.41 | 19.47 | 70.27 |

Table 7: Performance comparison of different models using different Triton block_size on the RULER dataset

| model | block size | 4k | 8k | 16k | 32k | 64k | 128k | avg |
|-------|-----------|------|------|------|------|------|------|------|
| GLM   | 128 | 93.51 | 91.83 | 89.90 | 91.35 | 86.06 | 83.41 | 89.34 |
|       | 64 | 94.23 | 92.07 | 89.90 | 91.11 | 86.06 | 82.69 | 89.34 |
| Yi    | 128 | 93.27 | 86.78 | 83.89 | 72.36 | 64.66 | 56.97 | 76.32 |
|       | 64 | 93.03 | 86.06 | 83.89 | 75.48 | 61.06 | 55.05 | 75.76 |
| Qwen  | 128 | 90.39 | 89.91 | 83.17 | 81.25 | 59.14 | 20.67 | 70.75 |
|       | 64 | 91.11 | 89.90 | 83.65 | 82.93 | 60.10 | 20.43 | 71.35 |

## F.3 REPRESENTATIVE QUERY SUBSET

We perform ablation experiments on the representative query subset used in our method. We replace the position of the subset used in sparse pattern determination and vertical-slash index selection from the sequence end to the middle. Results in Table 8 show that replacing the subset for sparse pattern determination has no significant effect on model performance. However, replacing the subset for vertical slash sparse index selection significantly reduces performance. This demonstrates the rationale for choosing the last $block\_size$ query vectors as the representative query subset.

Table 8: Performance comparison of the Llama-3.1-8B-Instruct model on the RULER dataset using different representative query set, where *last* denotes using the last $block\_size$ query vectors, and *middle* means using the middle $block\_size$ query vectors.

| model | $\hat{Q}$ (sparse pattern) | $\hat{Q}$ (vertical slash) | 4k | 8k | 16k | 32k | 64k | 128k | avg |
|-------|---------------------------|---------------------------|------|------|------|------|------|------|------|
| LLaMA | *last* | *last* | 95.43 | 93.51 | 94.71 | 89.42 | 82.93 | 79.09 | 89.18 |
|       | *middle* | *last* | 95.43 | 93.27 | 94.71 | 89.91 | 82.93 | 77.88 | 89.02 |
|       | *last* | *middle* | 68.27 | 51.68 | 38.46 | 40.87 | 19.23 | 10.58 | 38.18 |

## F.4 ALTERNATIVE IMPLEMENTATION FOR QUERY-AWARE INDEX SEARCH

In Algorithm 4, we flatten and normalize the estimated attention map, then identify the minimum computational budget required for the sum of scores to exceed a given threshold $\gamma$. We also explore an alternative implementation that performs query-wise index selection. In this approach, the cumulative attention scores of selected key blocks exceed $\gamma$ for each query block. As shown in Table 9, this alternative achieves comparable performance. We chose the global approach with the flatten

operation primarily for its implementation efficiency, as it simplifies the attention mechanism while maintaining performance parity.

Table 9: Performance comparison of different implementations for Query-Aware index search

| Implementation | LLaMA | GLM | Yi | Qwen |
|---|---|---|---|---|
| w/ flatten | 89.18 | 89.34 | 76.32 | 70.05 |
| wo/ flatten | 89.3 | 89.78 | 76.08 | 70.55 |

## F.5 MINIMUM BUDGET LIMIT

We conduct ablation experiments to analyze the effect of the minimum computational budgets. Table 10 demonstrates that when $\gamma$ is high, the model effectively captures most of the important tokens, making a minimum budget limit unnecessary. However, with smaller $\gamma$, the number of tokens selected by some attention heads may be insufficient, causing them to malfunction. In this scenario, implementing a minimum budget threshold significantly enhances the model's performance. Moreover, this minimum budget limit does not need to be excessive, as increasing the budget further does not lead to better results.

Table 10: Performance comparison of Llama-3.1-8B-Instruct and GLM-4-9B-Chat model using different minimum budget on the RULER dataset.

| model | $\gamma$ | min budget | 4k | 8k | 16k | 32k | 64k | 128k | avg |
|---|---|---|---|---|---|---|---|---|---|
| LLaMA | 0.95 | - | 94.23 | 93.27 | 93.03 | 92.31 | 85.34 | 78.37 | 89.42 |
| | 0.95 | 1024 | 95.43 | 93.51 | 94.71 | 89.42 | 82.93 | 79.09 | 89.18 |
| | 0.95 | 2048 | 95.19 | 93.27 | 94.23 | 89.90 | 82.93 | 77.64 | 88.86 |
| | 0.9 | - | 91.35 | 88.70 | 89.66 | 85.82 | 77.64 | 75.48 | 84.78 |
| | 0.9 | 1024 | 95.43 | 93.27 | 94.23 | 87.98 | 81.49 | 77.89 | 88.38 |
| | 0.9 | 2048 | 95.67 | 94.47 | 94.23 | 89.90 | 83.41 | 75.24 | 88.82 |
| GLM | 0.95 | - | 92.07 | 90.15 | 90.87 | 90.63 | 86.06 | 83.17 | 88.82 |
| | 0.95 | 1024 | 93.51 | 91.83 | 89.90 | 91.35 | 86.06 | 83.41 | 89.34 |
| | 0.95 | 2048 | 93.51 | 92.55 | 89.90 | 91.59 | 86.54 | 83.17 | 89.54 |
| | 0.9 | - | 86.78 | 83.17 | 87.50 | 89.66 | 83.17 | 81.25 | 85.26 |
| | 0.9 | 1024 | 93.75 | 91.59 | 89.42 | 90.63 | 83.90 | 82.21 | 88.58 |
| | 0.9 | 2048 | 93.51 | 92.07 | 89.66 | 91.35 | 84.62 | 81.73 | 88.82 |

## F.6 MAXIMUM BUDGET LIMIT

We perform ablation experiments to analyze the effect of the maximum computational budgets for each attention head on model performance and latency. Table 11 shows that capping the maximum computational budget affects different models variously. Imposing a maximum budget constraint on models like LLaMA leads to negative impacts. Conversely, for models like GLM, such a constraint maintains or even enhances performance. These findings highlight that different models are optimized for processing different context lengths.

## G LATENCY BREAKDOWN

The computational complexity of FlexPrefill comprises the following components:

- **Representative Attention Score Computation**: approximately $O(bnd)$, where $b$ is the block size, $d$ is the hidden dimension, and $n$ is the sequence length. This includes the computation of attention scores between the representative query set $\hat{Q} \in \mathbb{R}^{b \times d}$ and all key vectors $K \in \mathbb{R}^{n \times d}$.
- **Pattern Search**: approximately $O(bn)$, involving block-pooled attention score estimation and the calculation of the Jensen-Shannon divergence between the estimated and true attention distributions.

Table 11: Performance comparison of different models using different maximum budgets on the RULER dataset.

| model | max budget | 4k | 8k | 16k | 32k | 64k | 128k | avg |
|---|---|---|---|---|---|---|---|---|
| LLaMA | - | 95.43 | 93.51 | 94.71 | 89.42 | 82.93 | 79.09 | 89.18 |
| | 32k | 95.43 | 93.51 | 94.71 | 89.42 | 84.14 | 75.72 | 88.82 |
| | 16k | 95.43 | 93.51 | 94.71 | 89.42 | 84.37 | 72.12 | 88.26 |
| | 8k | 95.43 | 93.51 | 94.95 | 87.98 | 84.13 | 65.87 | 86.98 |
| GLM | - | 93.51 | 91.83 | 89.90 | 91.35 | 86.06 | 83.41 | 89.34 |
| | 32k | 93.51 | 91.83 | 89.90 | 91.35 | 86.54 | 84.14 | 89.54 |
| | 16k | 93.51 | 91.83 | 89.90 | 90.38 | 87.02 | 82.93 | 89.26 |
| | 8k | 93.51 | 91.83 | 91.11 | 90.87 | 86.06 | 82.45 | 89.30 |
| Yi | - | 93.27 | 86.78 | 83.89 | 72.36 | 64.66 | 56.97 | 76.32 |
| | 32k | 93.27 | 86.78 | 83.89 | 72.84 | 63.46 | 58.17 | 76.40 |
| | 16k | 93.27 | 86.78 | 83.65 | 71.63 | 62.74 | 57.69 | 75.96 |
| | 8k | 93.27 | 87.02 | 84.62 | 73.56 | 62.50 | 54.57 | 75.92 |
| Qwen | - | 90.39 | 89.91 | 83.17 | 81.25 | 59.14 | 20.67 | 70.75 |
| | 32k | 90.39 | 89.91 | 83.17 | 81.25 | 57.69 | 20.19 | 70.43 |
| | 16k | 90.39 | 89.91 | 83.17 | 81.97 | 59.86 | 21.15 | 71.07 |
| | 8k | 90.39 | 89.91 | 82.45 | 80.77 | 59.62 | 19.71 | 70.47 |

- **Sparse Index Construction**: approximately $O(n \log n)$, required for sorting the representative attention scores and constructing the sparse attention indices.

- **Sparse Attention Computation**: approximately $O(\alpha n^2 d)$, where $\alpha$ is the sparsity factor, representing the fraction of computations performed relative to dense attention.

In contrast, the standard dense attention mechanism has a computational complexity of $O(n^2 d)$. FlexPrefill introduces a modest overhead (approximately $O(\alpha n^2 d) + O(n \log n) + O(bnd)$), which is significantly offset by the computational savings achieved through sparsity.

Figure 6 presents the practical latency measurements for each component of FlexPrefill, including representative attention score computation, attention pattern search, sparse index construction, and sparse attention computation. At shorter input lengths, non-attention computation overheads are higher. As input length increases, index search construction time grows, but its percentage gradually decreases.

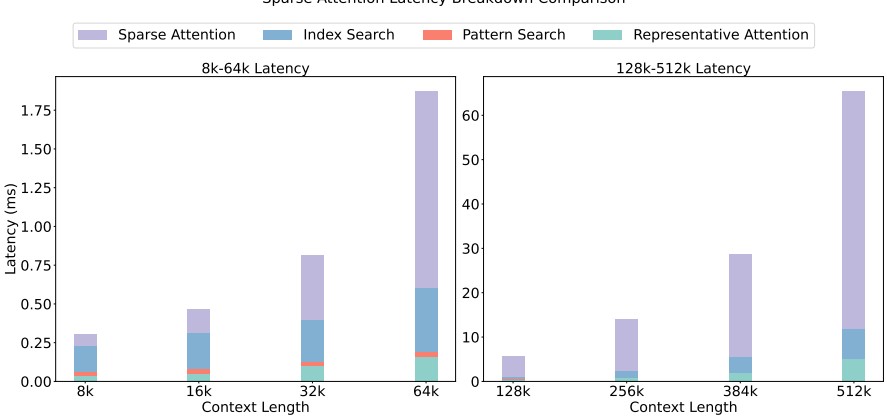

Figure 6: Sparse Attention Latency Breakdown Comparison across different context lengths. The graph shows the contribution of different components (Sparse Attention, Index Search, Pattern Search, and Representative Attention) to the overall latency for various input lengths. As input length increases, the proportion of time spent on sparse attention computation grows, while other components' relative contributions decrease.

## H   SPARSE PATTERN DISTRIBUTION

We analyze the Jensen-Shannon distance and sparse attention pattern distribution, focusing on *Query-Aware* head configurations. Figure 7 indicates that block-wise attention score estimation varies with task type and context length. Figure 8 shows most attention heads use *Vertical-Slash* patterns, with fewer *Query-Aware* patterns mainly in the model's first layer. These ablation studies provide insights into our method's components and configurations, enabling informed decisions to optimize sparse attention mechanism performance and efficiency.

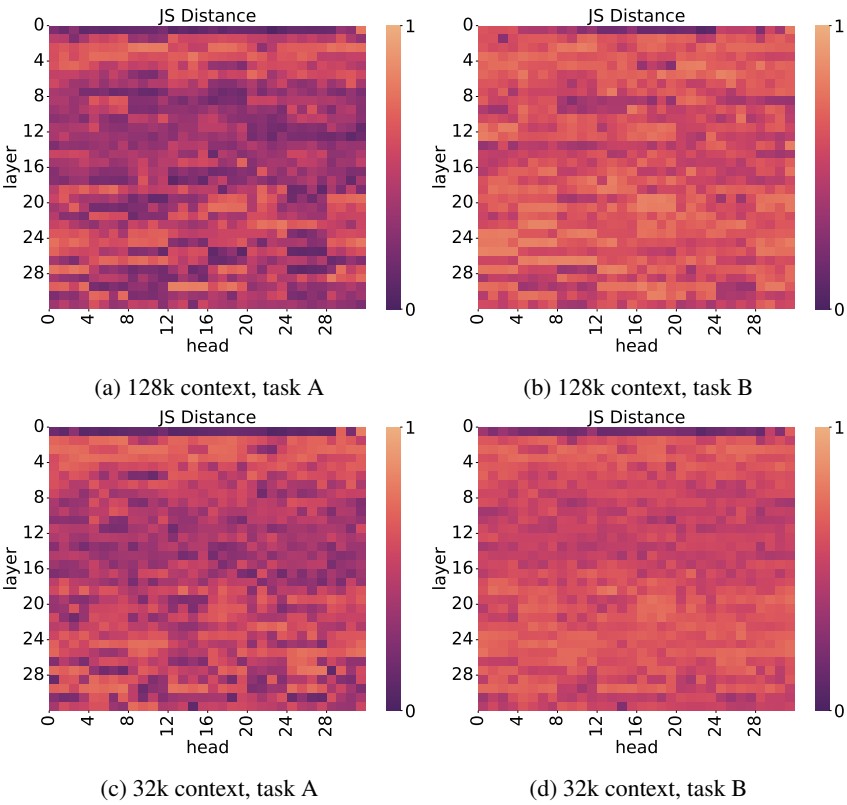

(a) 128k context, task A    (b) 128k context, task B

(c) 32k context, task A    (d) 32k context, task B

Figure 7: Jensen-Shannon (JS) distance heatmaps comparing sparse attention pattern distributions across different attention heads and layers. The comparison is shown for different context lengths (128k vs. 32k) and task types (task A vs. task B). Darker colors indicate lower JS distance, suggesting more accurate attention estimation of *Query-Aware* pattern.

## I   SPARSE ATTENTION MASK

The sparse patterns searched by our proposed algorithm for different attention heads are highly dynamic, and we visualize them on the Llama-3.1-8B-Instruct model. Figure 9 shows typical *Vertical-Slash* and *Query-Aware* attention heads, and demonstrates that there are large differences in the sparsity rates required for different attention heads.

## J   SPARSITY RATIO

We visualize the sparsity ratio of different samples on various attention heads in the Llama-3.1-8B-Instruct model. Figure 10 shows that samples with different difficulties require varying sparsity rates and have inconsistent sparsity distributions across attention heads. Additionally, different input lengths exhibit varying sparsity rates, with longer inputs showing higher sparsity ratios.

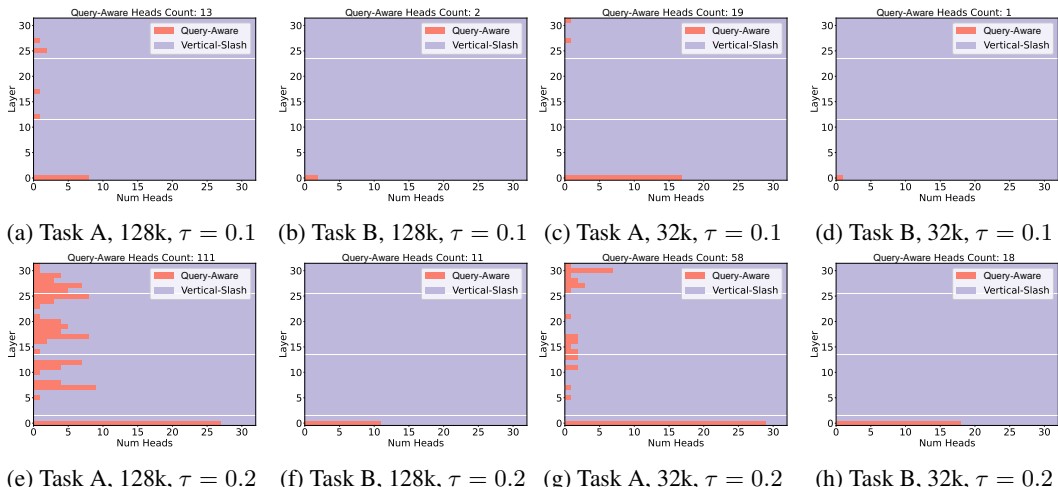

Figure 8: Comparison of the distribution of the number of different attention patterns between layers for different context lengths (128k vs. 32k) and task types (task A vs. task B).

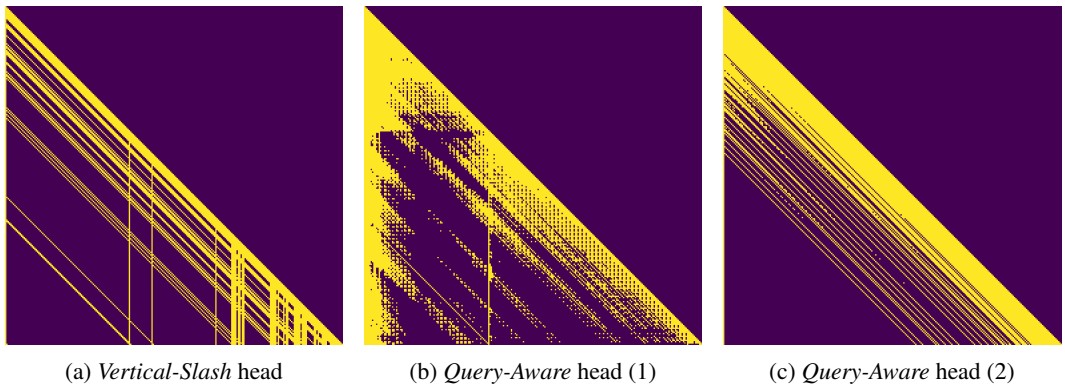

(a) *Vertical-Slash* head     (b) *Query-Aware* head (1)     (c) *Query-Aware* head (2)

Figure 9: Visualization of sparse masks for different attention heads in the Llama-3.1-8B-Instruct model. (a) shows the sparse mask of *Vertical-Slash* heads. (b) shows the sparse mask of *Query-Aware* head, where there are many diverse blocks that stray from a specific pattern. (c) shows that a *Query-Aware* head may still exhibit the *Vertical-Slash* pattern.

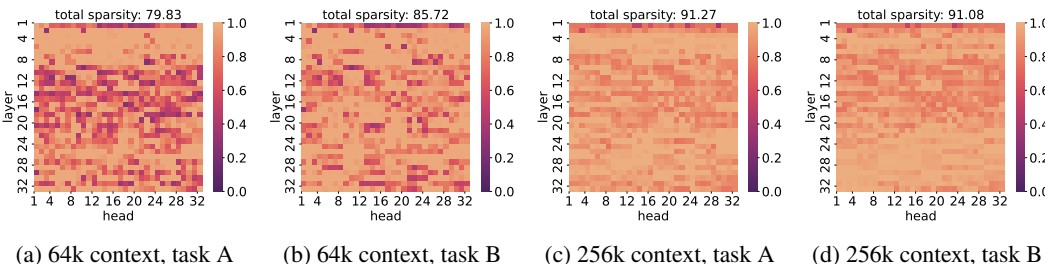

(a) 64k context, task A    (b) 64k context, task B    (c) 256k context, task A    (d) 256k context, task B

Figure 10: Visualization of sparsity ratios across different attention heads in the Llama-3.1-8B-Instruct model. The heatmaps show varying sparsity distributions for different sample types (task A vs. task B) and context lengths (64k vs. 256k). Darker colors indicate lower sparsity. Longer inputs (c, d) exhibit higher overall sparsity ratios compared to shorter inputs (a, b).

