# OpenReview forum: "FlexPrefill: A Context-Aware Sparse Attention Mechanism for Efficient Long-Sequence Inference"
_ICLR.cc/2025/Conference — ICLR 2025 Oral_

### Official Review · Reviewer_mNvJ · 2024-10-26

**Soundness:** 3
**Presentation:** 3
**Contribution:** 3
**Rating:** 8
**Confidence:** 3

**Summary:**

This work investigates the LLM long sequence inference efficiency challenge and proposes a novel and flexible method FlexPrefill to solve it. The proposed FlexPrefill featured with two components: Query-Aware Sparse Pattern Determination and Cumulative-Attention Based Index Selection, of which the former one adaptively optimize the usage of the sparse attensions while the latter one ensures the quality of selected sparse attentions. Extensive experiments elaborate the effectiveness and superiority of FlexPrefill.

**Strengths:**

a) The proposed Query-Aware Sparse Pattern Determination strategy step further on existing work to provide flexibility to sparse attention usage.

b) Promising improvments on extensive experiment compared with SOTA.

**Weaknesses:**

a) Unclear costs analysis of pre-calculation on JS divergence.

b) The baseline selections are insufficient.

**Questions:**

a) Could the authors elaborate the intuition behind the usage of JS divergence? Would the W-Distence be more appropriate since W-Distence help measures the change costs of given two attentions? Also, how the JS divergence can help FlexPrefill to be effective and flexible, would it introduce addtional calculation costs?

b) Do the authors have a vision on how the model structure can impact on the performance of FlexPrefill?

c) For the baseline selections, concerns may arise since the FlexPrefill poses an automatic selection process which makes FlexPrefill more flexible while the baselines do have. It would be better to see the comparisons with methods equipped automatic selections such as Moa[1].

d) In formula (2), should the JS distance be the calculation on a||b and b||a given two distributions a, b. Can the authors elaborate the usage of a\hat{-}||m and a\hat{^}||m?


**Reference**

[1] Moa: Mixture of sparse attention for automatic large language model compression.

---

> ### Author Response · Authors · 2024-11-19
> **Response to Reviewer mNvJ**
>
> **R W1: Unclear Cost Analysis of Pre-calculation on** **JS Divergence**
>
> We note that the pre-calculation of JS divergence involves the following computational costs:
>
> - **Representative Attention Score Computation**: $O(bnd)$, where $b$ is the block size, $d$ is the head dimension, and $n$ is the sequence length.
> - **JS Divergence Computation**: $O(bn)$, including block-wise attention distribution estimation and JS divergence computation.
>
> We will incorporate these detailed cost analyses into our next revision to provide a clearer and more comprehensive representation.
>
>
> **R Q1:** **JS Divergence** **Usage and Efficiency**
>
> While the [Wasserstein distance](https://en.wikipedia.org/wiki/Wasserstein_metric) is a potential alternative, we selected [Jensen-Shannon divergence](https://en.wikipedia.org/wiki/Jensen%E2%80%93Shannon_divergence) (JS divergence) for both theoretical and practical reasons. JS divergence is computationally efficient and provides bounded (0 to 1), symmetric measurements, making it well-suited for our application.
>
> We use JS divergence to evaluate whether block-wise attention estimation sufficiently approximates the true distribution. This enables flexible switching between query-aware and vertical-slash patterns. As shown in Figure 6, the additional computational cost introduced by the pattern search component is minimal, contributing less than 5% to the total latency.
>
>
> **R Q2: Model Structure Impact**
>
> FlexPrefill has demonstrated robust performance across diverse model architectures, including Llama, GLM, Yi, and Qwen, as shown in our experimental results. This consistent performance across different architectures suggests that our method's adaptive nature allows it to effectively accommodate various model structures.
>
>
> **R W2: insufficient baselines & Q3: Automatic Selection Comparison**
>
> While automatic selection methods like MoA are interesting, we focused our comparison with MInference as it represents a strong baseline. Moreover, we conducted additional experiments comparing FlexPrefill with MoA across different sequence lengths:
>
> For Llama-3-262k-instruct:
>
> |             | 4k    | 8k    | 16k   | 32k   | 64k   | 128k  | Avg   | SpeedUp (128k) |
> | ----------- | ----- | ----- | ----- | ----- | ----- | ----- | ----- | -------------- |
> | FlexPrefill | 90.87 | 89.9  | 90.38 | 82.21 | 81.25 | 79.81 | 85.74 | 2.43x          |
> | MoA         | 91.35 | 88.22 | 85.58 | 83.9  | 81.97 | 76.44 | 84.58 | 1.31x          |
>
> These results demonstrate that FlexPrefill not only achieves better average performance (85.74% vs 84.58%) but also provides speedup (2.43x vs 1.31x at 128k tokens). FlexPrefill's dynamic pattern selection operates efficiently while delivering better performance-speed trade-offs.
>
>
> **R Q4: JS Distance Formulation**
>
> Our implementation follows the original [Jensen-Shannon divergence definition](https://en.wikipedia.org/wiki/Jensen%E2%80%93Shannon_divergence) using the mean distribution $m=(\bar{a}+\hat{a})/2$.
>
> While [symmetrised KL divergence](https://en.wikipedia.org/wiki/Kullback%E2%80%93Leibler_divergence#Symmetrised_divergence) could also measure distribution distances, we chose JS divergence for its symmetry and boundedness properties, which provide more stable pattern determination across different attention scenarios

---

> > ### Comment · Reviewer_mNvJ · 2024-11-19
> > **Thank you for the response**
> >
> > Thank you for the detailed explainations to my concerns. It's more clear to me now. It would be better to see how JS is better than Wasserstein distance from both theoretical and practical aspects.
> > I decide to raise my ratings, please include our disccusions into the revision.

---

### Official Review · Reviewer_3fJR · 2024-11-03

**Soundness:** 3
**Presentation:** 3
**Contribution:** 2
**Rating:** 8
**Confidence:** 4

**Summary:**

This paper addressed the flexibility of the sparse attention pattern to various input length and task currently applied in LLM. The authors split the attention heads into two types: the Query-Aware heads that capture variable sparse pattern upon query, and the Vertical-Slash heads that capture structured and invariant sparse patterns observed in pervious works. The authors decide the sparse pattern for each head by evaluating the discrepancy between the estimated and true attention distribution, and select the sparse pattern by summing up the largest attention scores to a threshold. The proposed method balances the computational efficiency with attention accuracy. The authors evaluate their approach on RULER and Infinite Bench and show that their method preserves high performance across multiple context length and tasks.

**Strengths:**

- The paper propose a more dynamic and flexible approach for sparse attention calculation. Their method performs sparse pattern search and sparse index selection both in a dynamic, query-aware manner. The flexibility and dynamism distinct it from MInference, and contribute to the performance improvement.
- The method is based on solid observations.
- The paper gives a concrete and simple algorithm to construct these sparse attention patterns.
- The "Cumulative-Attention Based Index Selection" method is convincing with the details discussed in Appendix A. The authors derive the algorithm from the original  multi-objective optimization problem in details, providing the rationale in summing up the largest attention scores.

**Weaknesses:**

Methodology:

- **Insufficient Comparison with Baselines**: There are also other sparse attention method in recent works such as SampleAttention [1] and HiP-Attention [2]. The drawbacks of StreamingLLM in handling longer contexts is obvious in other papers, it would be better to compare to other more recent methods.
- **Insufficient evaluation of latency**: Although "Cumulative-Attention Based Index Selection" improves performance, it may result in unbalanced computation allocation between different attention heads. I think this could lead to an overall increase in latency, as MInference explicitly avoids this by setting a target FLOPs for each pattern. I think it is necessary to show to what extend the dynamic, head-distinctive computation allocation increases the end-to-end latency compared to the fixed budget setting within the same FLOPs.
- **Determination of attention patterns**: The algorithm select the vertical-slash pattern when the distribution of estimated and true attention is not that close. However, in my opinion, the distribution discrepancy only indicates that the result of block-wise sparse attention deviates the full attention, but doesn't lead to using the vertical-slash patterns. Is there any theoretical or empirical guarantee that vertical flash patterns are better than block-wise pattern in this scenario?

Presentation:

- "eacct" -> "exact" in Line 1,043.
- The caption of Table 5 should be above the table.
- In Figure 4, it would be more clear to show the values of $\gamma$ for each point, which will be helpful to provide an empirical selection of $\gamma$.
- The authors states "accelerating the computational process" in Line 357 but Table 1 does not include any result concerning efficiency. It would be better to give a reference to figure 4 here.

[1] Zhu, Q., Duan, J., Chen, C., Liu, S., Li, X., Feng, G., ... & Yang, C. (2024). Near-lossless acceleration of long context llm inference with adaptive structured sparse attention. *arXiv preprint arXiv:2406.15486*.

[2] Lee, H., Park, G., Lee, Y., Kim, J., Jeong, W., Jeon, M., & Hwang, S. J. (2024). HiP Attention: Sparse Sub-Quadratic Attention with Hierarchical Attention Pruning. *arXiv preprint arXiv:2406.09827*.

**Questions:**

- Q1: Figure4 shows the attention latency and performance on RULER. However, this figure is not comprehensive enough. Why there is only one point for MInference? Why not adjust the computation budget of MInference and plot a curve like "Ours"? A curve for "Fixed Budget Query-Aware" is also missing.
- Q2: In Algorithm 4, the attention map is normalized among all the attention scores, but Algorithm 3 only computes the mean values on each vertical and slash lines. The mean operation in Algorithm 3 can not assure the sum of $a_v$ and $a_s$ to be 1, thus the criterion of Algorithm 3 and 4 may be different even with the same $\gamma$. Could the authors explain why there is no normalization in Algorithm 3?
- Q3: in Appendix A, Step 7, the equivalent form of the dual objective is not that intuitive. How can we derive this from the equation in Step 6?

---

> ### Author Response · Authors · 2024-11-19
> **Title: Response to Reviewer 3fJR (1/2): Response to the Weaknesses Part**
>
> **R W1: Insufficient Baseline Comparisons:**
>
> We here provide a direct comparison with HiP-Attention using results from their original paper for Llama 3.1 8B:
>
> | Method      | 4k    | 8k    | 16k   | 32k   | 64k   | 128k  | Avg   | SpeedUp (128k) |
> | ----------- | ----- | ----- | ----- | ----- | ----- | ----- | ----- | -------------- |
> | FlexPrefill | 95.43 | 93.51 | 94.71 | 89.42 | 82.93 | 79.09 | 89.18 | 2.4x           |
> | HiP         | 96    | 94.7  | 94.1  | 89.7  | 79.9  | 58.2  | 85.4  | 1.7x           |
>
> This comparison demonstrates that FlexPrefill achieves both better average performance (89.18% vs 85.4%) and better computational efficiency (2.4x vs 1.7x speedup at 128k tokens). Notably, FlexPrefill maintains significantly better performance on longer sequences, with a 20.89 percentage point advantage at 128k tokens, while also achieving higher speedup.
>
> We note that the implementation of SampleAttention is not publicly available at this time. Once it becomes accessible, we will incorporate it into our comparison for a more comprehensive evaluation.
>
>
> **R W2: Concerns about potential latency increase from unbalanced computation allocation across attention heads.**
>
> We conducted comprehensive measurements for Llama 3.1 8B with 128k context and 32 heads to analyze this impact. The following experiment used the same average computational budget (11%).
>
> | Configuration | Balanced Budget | Unbalanced Budget | Latency Overhead |
> | ------------- | --------------- | ----------------- | ---------------- |
> | Time (ms)     | 117.75          | 119.89            | 2.14             |
>
> We note that **Unbalanced Budget's** computational budget was significantly unbalanced, varied from 1.74% to 43.33% across attention heads.
>
> Our empirical measurements demonstrate that the practical impact on latency is minimal. Our experiments show that the additional latency overhead is quite modest - approximately 2ms per attention call compared to balanced budget approaches.
>
>
> **R W3: Questions the justification for defaulting to vertical-slash patterns when attention distribution estimation shows a discrepancy.**
>
> Our choice of using vertical-slash patterns as a fallback is motivated by empirical observations of attention patterns in large language models, as illustrated in Figure 2 of our paper. Through extensive analysis, we observed that attention heads predominantly exhibit either Query-Aware or Vertical-Slash patterns.
>
> While we acknowledge that the distribution discrepancy alone doesn't theoretically necessitate the use of vertical-slash patterns, as stated in line 199 of our paper, in such cases, "we fall back to a more conservative approach", our empirical results support this as a robust fallback strategy that maintains model performance while ensuring computational efficiency.

---

> > ### Public Comment · ~Xiao_Chuanfu1 · 2024-11-27
> >
> > We read FlexPrefill with great interest and appreciate its contribution to sparse attention research. We would like to bring to the authors' and reviewers’ attention one prior work, SampleAttention (https://arxiv.org/abs/2406.15486, June 2024), which appears to share some foundational concepts with FlexPrefill.
> > Given the temporal precedence of SampleAttention and the significant conceptual overlap, specifically:
> > 1. The core observations regarding Dynamic Sparse Pattern and Adaptive Sparsity Ratio
> > 2. The methodological approaches of "Query-Guided Attention Sampling and Score-Based Key-Value Filtering" in SampleAttention, which appear closely related to FlexPrefill's "Query-Aware Sparse Pattern Determination and Cumulative-Attention Based Index Selection"
> >
> > We believe it would strengthen the paper's contribution to include a clear delineation of FlexPrefill's novel contributions relative to existing work, especially SampleAttention.
> >
> > This addition would help readers better understand FlexPrefill's unique advances in the field of adaptive sparse attention. We would be happy to provide more detailed technical comparisons if helpful.
> >
> > We look forward to seeing how the authors position their valuable work within the existing literature.

---

> ### Author Response · Authors · 2024-11-19
> **Response to Reviewer 3fJR (2/2): Response to the Questions Part**
>
> **R Q1: Question about the comprehensiveness of Figure 4 and comparison curves**
>
> Figure 4 demonstrates our method's key advantage of flexible performance-latency trade-offs through $\gamma$ adjustment. The single point for MInference reflects a practical limitation - as shown in Table 7 of the [MInference (NeurIPS 2024)](https://arxiv.org/abs/2407.02490) paper, their method only supports six predefined combinations of patterns and ratios, each requiring extensive offline search to determine optimal configurations. We selected their paper's recommended configuration as a representative comparison point. In contrast, FlexPrefill offers continuous adjustment through $\gamma$, enabling fine-grained control over the performance-latency trade-off without requiring additional offline search processes.
>
> We have chosen not to include the performance of "Fixed Budget Query-Aware" due to its significantly degraded results. Notably, its poorer performance underscores the importance of our pattern selection mechanism, which is a key innovation of our FlexPrefill method.
>
>
> **R Q2: Question about normalization differences between Algorithms 3 and 4**
>
> The apparent difference in normalization between Algorithms 3 and 4 represents two mathematically equivalent implementations of the same concept. While Algorithm 4 explicitly shows normalization across all attention scores, Algorithm 3's mean operation over vertical and slash lines achieves the same effect through a different computational path.
>
> We plan to revise the algorithm descriptions to better highlight this equivalence in the final version.
>
>
> **R Q3: The rationale behind the** **transition from Step 6 to Step 7** The transition from Step 6 to Step 7 follows from the properties of the Lagrangian dual problem. We now provide the key intuition:
>
> In Step 6, the objective maximizes over λ with the expression:
> $\sum\limits(\exp(x_{j})/\sum\limits \exp(x_j')-\lambda)+\lambda\gamma_S$
> for all $j$ where $\exp(x_j)/\sum\limits \exp(x_j')\ge\lambda$
>
> For any fixed $\lambda$, this naturally defines a set $S$ of indices where $\exp(x_j)/\sum\limits\exp(x_j') \ge \lambda$. The optimal $\lambda^*$ will be the smallest value that satisfies the constraint in Step 7, because any larger value would reduce the objective in Step 6. Therefore, minimizing $|S|$ subject to the sum constraint in Step 7 is equivalent to finding the optimal $\lambda^*$ in Step 6.
>
> The relationship $\gamma_a = 1 - \lambda^*\gamma_S$ emerges from complementary slackness in the KKT conditions, where $\lambda$ represents the optimal trade-off between set size and attention score coverage.
>
> We will add more detailed intermediate steps in the final version to make this derivation more transparent.
>
> **R Presentation:**
>
> We express our sincere gratitude to your insightful comments. Regarding the presentation issues raised, we will carefully incorporate all suggested improvements in our next revision.

---

> > ### Comment · Reviewer_3fJR · 2024-11-24
> > **Thank you for the response**
> >
> > Thank you for your thorough response. Your explanation has addressed most of my concerns. I will raise my rating accordingly.

---

> ### Author Response · Authors · 2024-11-27
>
> Thank you for your interest and acknowledgment to our work.
>
>
>
> After carefully reviewing the SampleAttention [1] paper, we would like to clarify the distinctions between our works. We have positioned these observations in Section 2 of our paper (Lines 119-141), specifically in paragraphs "Attention Sparse Patterns Exhibit Variability" and "Varying Samples Require Adaptive Sparsity Ratio", as motivational context rather than claiming them as novel discoveries. We acknowledge that other sparse attention works, including MInference and SampleAttention, were similarly motivated by these phenomena, and **we will include comprehensive references to provide a complete background.**
>
>
>
> Regarding methodology, it is important to note that our "Query-Aware Sparse Pattern Determination" differs fundamentally from "Query-Guided Attention Sampling."
>
> - Our term "Query-Aware" refers to a specific pattern defined in lines 174-175, **which is designed to capture variable sparse patterns contingent upon query positions.**
>
> - "Query-Aware Pattern Determination" in our work involves applying the last query block to analyze differences between block-wise attention distributions (as detailed in lines 184-185), **which then informs our selection of the appropriate sparse attention pattern.**
>
>
> This approach stands in clear contrast to SampleAttention's methodology, where "Query-Guided" refers to their KV selection process. Specifically, SampleAttention uses query guidance primarily for selecting key-value indices, computing exact scores for selected queries in Stage-1 to determine KV indices of interest in Stage-2. This fundamental difference in purpose and implementation distinguishes our methodological contributions.
>
>
>
> While we share similar goals for sparse attention, our implementation differs significantly.
>
> - We first determine patterns before KV selection, as we employ **different query-key pair selection strategies across patterns, heads, and inputs.** This contrasts with SampleAttention's fixed "local window pattern" and "column stripe pattern" approach for all heads.
>
> - Furthermore, while SampleAttention samples a subset of queries to minimize CRA, we apply different strategies for different heads.
>
>     - For "query-aware head" we utilize **all queries with a pooling operation** to estimate the attention map and select **different Q-K pairs for different queries**.
>
>     - For the remaining heads, we apply the last block query to estimate vertical and slash lines.
>
>
> For a clear understanding of the differences between the two approaches, we encourage readers to compare **Figure 9 in FlexPrefill with Figure 1 in SampleAttention**, which visually illustrates the distinct sparse attention mask of each method.
>
>
>
> **We uploaded our implementation code to the supplementary materials** and highlighted that, a key distinction of FlexPrefill lies in its flexibility. FlexPrefill can perform online pattern determination and query-key index selection without offline search. This allows FlexPrefill to be integrated into existing LLM frameworks through a single function call replacing standard attention mechanisms, such as in vLLM and transformers.
>
>
>
> Regarding benchmarking, we acknowledge that we haven't made direct comparisons with SampleAttention due to its implementation not being publicly available. We would welcome guidance on implementation details to ensure accurate reproduction and incorporate it into our comparative evaluation.
>
>
>
> [1] [SampleAttention](https://arxiv.org/abs/2406.15486)

---

### Official Review · Reviewer_9EQr · 2024-11-04

**Soundness:** 3
**Presentation:** 4
**Contribution:** 3
**Rating:** 8
**Confidence:** 3

**Summary:**

The paper introduces FlexPrefill, a novel sparse attention mechanism aimed at improving the efficiency of long-sequence inference for large language models (LLMs). The proposed approach addresses the computational challenges associated with long-sequence processing by dynamically adjusting sparse attention patterns and computational budgets based on input characteristics. FlexPrefill consists of two key components: Query-Aware Sparse Pattern Determination, which selects between diverse and structured attention patterns using Jensen-Shannon divergence, and Cumulative-Attention Based Index Selection, which ensures efficient attention computation by dynamically selecting query-key indices based on attention scores. Experimental results demonstrate  improvements in both speed and accuracy compared to existing methods, such as Streaming LLM and Minference.

**Strengths:**

1. Originality: The paper presents an adaptive and flexible approach to sparse attention that addresses the limitations of fixed sparse attention mechanisms. The real-time adjustment of sparse attention patterns is a notable advancement in handling varying input needs during long-sequence inference, contributing to the growing body of research focused on optimizing LLM efficiency.

2. Quality: The quality of the experimental setup is high, with comprehensive comparisons to baseline methods like Streaming LLM and Minference. The authors provide detailed analyses that demonstrate improvements in both speed and accuracy, supporting the efficacy of the proposed method. The ablation studies further strengthen the validity of the claims by examining the contributions of each component of the approach.

3. Clarity: The paper is well-structured and clearly written, making it accessible to a broad audience. The authors provide a thorough explanation of the motivation, methodology, and experimental results.

**Weaknesses:**

1. Complexity of the Proposed Sparse Attention Mechanism: The multi-step decision-making process, involving both Query-Aware Sparse Pattern Determination and Cumulative-Attention Based Index Selection, introduces additional complexity to the attention mechanism. A detailed computational complexity analysis is missing, which raises questions about how the overhead introduced by this process is justified by the computational savings achieved through sparse attention.

2. Lack of Detailed Comparison of Computational Savings: The paper lacks a comprehensive comparison of how much computation is saved by using FlexPrefill compared to the baselines presented in Tables 1 and 2. This information would be valuable in quantifying the benefits of the proposed approach.

3. Limited Comparison with Recent Related Works: The paper does not include comparisons with other recent related works, such as Han et al. (2024) and Xiao et al. (2024a). Including these comparisons would help to better position the contributions of FlexPrefill within the context of current research.

4. Mixed Performance Benefits: The reported performance benefits compared to the Full Attention model on long-context benchmarks are mixed, as shown in Tables 1 and 2. Additionally, the computational savings compared to the Full Attention model and other baselines are not adequately reported, which limits the understanding of the trade-offs involved.

**Questions:**

1. Can you provide a detailed computational complexity analysis on the proposed sparse attention mechanism?

2. Given the complexity of the multi-step decision-making process, how feasible is it to integrate FlexPrefill into existing LLM frameworks without significant modifications?

3. In Figure 4, what is the average attention latency of the Full Attention model?

4. Can you include a comparison of how much computation is saved by using FlexPrefill compared to the baselines in Tables 1 and 2?

5. Have you considered including direct comparisons with other recent related works, such as Han et al. (2024) and Xiao et al. (2024a)? How does FlexPrefill compare to these methods?

6. How were the values of the thresholds (e.g., sparse pattern threshold and cumulative attention threshold) chosen? Were these values empirically determined, or based on specific criteria?

---

> ### Author Response · Authors · 2024-11-19
> **Title: Response to Reviewer 9EQr (1/3)**
>
> **R W1: Detailed computational complexity analysis to justify overhead of multi-step sparse** **attention mechanism** **& Q1: Computational Complexity Analysis.**
>
> We here provide a clear justification for the overhead associated with FlexPrefill's multi-step sparse attention mechanism. As outlined in **Figure 6** of our paper, the analysis spans context lengths ranging from 8K to 512K tokens and demonstrates that the incurred overhead is minimal compared to the computational savings achieved.
>
> 1. **Computational Complexity Breakdown**
>
> The computational complexity of FlexPrefill is composed of the following components:
>
> - **Representative Attention Score Computation**: $O(bnd)$, where $b$ is the block size, $d$ is the hidden dimension, and $n$ is the sequence length.
> - **Pattern Search**: $O(bn)$, which involves computing JS divergence and dynamically selecting the most suitable attention pattern.
> - **Sparse Index Construction**: $O(n \log n)$, required for efficiently organizing sparse attention indices.
> - **Sparse Attention Computation**: $O(\alpha n^2 d)$, where $\alpha$ represents the sparsity ratio.
>
> In contrast, the standard attention mechanism has a computational cost of $O(n^2 d)$. By comparison, FlexPrefill introduces a small overhead (approximately $O(\alpha n^2 d)+O(n\log n)$), which is substantially outweighed by the computational savings enabled by the sparsity mechanism.
>
> 2. **Empirical Justification for Long Sequences**
>
> Our empirical evidence further supports the efficiency of FlexPrefill, particularly for long sequences ($n \geq 128k$). As demonstrated in **Figure 6**, the overhead becomes increasingly negligible as the dominant computation shifts to sparse attention.
>
>
> **R W2: Need for more detailed quantification of computational savings compared to baselines & Q4: Computational Savings.**
>
> We conducted comprehensive experiments to quantify computational costs across four LLMs, measuring both the average time of a single attention computation and accuracy:
>
> For Llama:
>
> | Llama          | 64k time (ms) | 128k time (ms) | average time (ms) | RULER score |
> | -------------- | ------------- | -------------- | ----------------- | ----------- |
> | Full Attention | 157.88        | 658.63         | 144.97            | 88.7        |
> | Streaming LLM  | 88.41         | 180.4          | 56.08             | 61.62       |
> | Minference     | 152.56        | 215.53         | 116.63            | 85.42       |
> | FlexPrefill    | 85.4          | 271.07         | 70.61             | 89.18       |
>
> For GLM:
>
> | GLM            | 64k time (ms) | 128k time (ms) | average time (ms) | RULER score |
> | -------------- | ------------- | -------------- | ----------------- | ----------- |
> | Full Attention | 267.26        | 1066.21        | 237.65            | 89.06       |
> | Streaming LLM  | 88.1          | 179.95         | 55.72             | 65.91       |
> | Minference     | 295.95        | 493.12         | 226.71            | 89.18       |
> | FlexPrefill    | 120.07        | 408.74         | 100.75            | 89.34       |
>
> For Yi:
>
> | Yi             | 64k time (ms) | 128k time (ms) | average time (ms) | RULER score |
> | -------------- | ------------- | -------------- | ----------------- | ----------- |
> | Full Attention | 243.17        | 1018.07        | 223.53            | 79.49       |
> | Streaming LLM  | 107.45        | 218.96         | 68.4              | 61.22       |
> | Minference     | 171.48        | 305.69         | 138.86            | 76          |
> | FlexPrefill    | 102.32        | 319.43         | 83.3              | 76.32       |
>
> For Qwen:
>
> | Qwen           | 64k time (ms) | 128k time (ms) | average time (ms) | RULER score |
> | -------------- | ------------- | -------------- | ----------------- | ----------- |
> | Full Attention | 137.53        | 569.86         | 125.67            | 68.83       |
> | Streaming LLM  | 77.62         | 157.51         | 49.46             | 52.96       |
> | Minference     | 220.51        | 306.47         | 162.25            | 66.07       |
> | FlexPrefill    | 61.51         | 173            | 47.62             | 70.75       |
>
> Our results reveal computational advantages while maintaining or improving performance.
>
> At 64k tokens, FlexPrefill demonstrates better efficiency compared to MInference, which shows higher latency than full attention. For Llama, FlexPrefill achieves 85.40ms latency compared to MInference's 152.56ms while maintaining higher accuracy (89.18% vs 85.42%).
>
> In long-sequence scenarios (128K tokens), FlexPrefill continues to show performance-efficiency trade-offs. For Qwen, we achieve 173ms latency while maintaining 70.75% accuracy, compared to MInference's 306.47ms.
>
> Across all sequence lengths, FlexPrefill consistently demonstrates practical speed advantages while maintaining or improving model performance.

---

> ### Author Response · Authors · 2024-11-19
> **Title: Response to Reviewer 9EQr (2/3)**
>
> **R W3: Limited comparison scope with recent works in sparse attention & Q5: Comparisons with Recent Works.**
>
> We appreciate this suggestion and have conducted additional experiments incorporating recent baselines. Below are comprehensive comparison results on the RULER dataset (We note that LM-Infinite does not support GLM and Yi):
>
> |              |             | 4k    | 8k    | 16k   | 32k   | 64k   | 128k  | avg   |
> | ------------ | ----------- | ----- | ----- | ----- | ----- | ----- | ----- | ----- |
> | Llama-3-262k | FlexPrefill | 90.87 | 89.9  | 90.38 | 82.21 | 81.25 | 79.81 | 85.74 |
> |              | MInference  | 91.35 | 88.46 | 87.74 | 83.65 | 81.73 | 77.88 | 85.14 |
> |              | InfLLM      | 89.4  | 79.8  | 70.1  | 55.6  | 43    | 39.5  | 62.9  |
> |              | LM-Infinite | 91.11 | 54.81 | 38.94 | 21.39 | 14.66 | OOM   | -     |
> |              |             |       |       |       |       |       |       |       |
> | GLM          | FlexPrefill | 93.51 | 91.83 | 89.9  | 91.35 | 86.06 | 83.41 | 89.34 |
> |              | Minference  | 93.75 | 93.03 | 90.38 | 89.9  | 85.1  | 82.93 | 89.18 |
> |              | InfLLM      | 94.7  | 89.5  | 76.4  | 66.5  | 56.8  | 53.5  | 72.9  |
> |              |             |       |       |       |       |       |       |       |
> | Yi           | FlexPrefill | 93.27 | 86.78 | 83.89 | 72.36 | 64.66 | 56.97 | 76.32 |
> |              | Minference  | 92.79 | 85.58 | 82.69 | 73.32 | 63.94 | 57.69 | 76    |
> |              | InfLLM      | 80.3  | 83.9  | 60.7  | 45.2  | 38.6  | 30.2  | 56.5  |
>
> Our choice of MInference as a primary baseline was deliberate. As demonstrated in their work, MInference already outperforms many recent methods in similar evaluation settings. By showing efficiency improvements over MInference, our method implicitly demonstrates advantages over these other approaches.
>
>
> **R W4: Questions about performance benefits and computational trade-offs compared to Full Attention and baselines.**
>
> Our experimental results demonstrate that FlexPrefill achieves comparable or better performance than both Full Attention and MInference while providing computational advantages.
>
> As shown in Table 1 in our original paper, for the Llama model, FlexPrefill maintains an average accuracy of 89.18% compared to Full Attention's 88.70% and MInference's 85.42%. This performance is achieved while delivering speedups. Similar patterns are observed across other models.
>
> We acknowledge the importance of providing a detailed comparison of computational costs. Please refer to **R W2** for further details.

---

> ### Author Response · Authors · 2024-11-19
> **Title: Response to Reviewer 9EQr (3/3)**
>
> **R Q2: Integration Feasibility**
>
> FlexPrefill is designed for seamless integration into existing LLM frameworks through a single function call that can replace standard attention mechanisms. This design makes it readily compatible with popular frameworks like transformers and vLLM. Unlike MInference, which requires model-specific pattern search, our method provides a unified interface that works consistently across different models and scenarios.
>
>
> **R Q3: Full Attention** **Latency**
>
> In Figure 4, the average latency of the full attention model is reported as follows: 4.5 ms (averaged across all input lengths), 20.58 ms (for 128k input length), and 1.19 ms (for 32k input length).
>
> Our proposed method, as illustrated in Figure 4, demonstrates varying latency depending on the parameter $\gamma$, while maintaining comparable performance. Specifically, our method achieves improved latency: 1.7 ms (averaged across all input lengths), 6 ms (for 128k input length), and 0.9 ms (for 32k input length).
>
> We will include a detailed side-by-side comparison with Flexprefill in our next revision to provide a clearer comparison and further highlight the efficiency of our approach.
>
>
> **R Q6: Threshold Selection**
>
> Our threshold parameters offer flexible control over the speed-accuracy trade-off. The cumulative attention threshold $\gamma$ allows direct tuning of the computation-performance balance - higher $\gamma$ values favor accuracy while lower values prioritize speed. The sparse pattern threshold $\tau$ was empirically determined. This flexibility enables users to adjust parameters based on their computational resources and performance requirements.

---

> ### Comment · Reviewer_9EQr · 2024-11-22
>
> Thank you for addressing my concerns! I will update my rating accordingly.

---

### Official Review · Reviewer_35xX · 2024-11-04

**Soundness:** 3
**Presentation:** 3
**Contribution:** 3
**Rating:** 8
**Confidence:** 4

**Summary:**

This paper introduces a pre-filling mechanism for LLMs, FlexPrefill, which dynamically adjusts sparse attention patterns and computational budget with respect to different input examples. There are two key innovations:
- Question aware sparse pattern determination, which leverage Jensen-Shannon divergence to determine whether an attention head should apply the "question-aware" pattern with respect to the given query. If not, it will apply the "vertical-slash" pattern similar to MInference.
- Cumulative-attention based index selection, which selects the sparse index by ensuring the sum of the normalized attention scores meets a predefined cumulative attention threshold, rather than selecting the sparse index to meet a given budget.

Experimental results indicate that FlexPrefill achieves comparable results with previous SOTA methods with GLM and Yi, and outperforms previous SOTA methods with a large margin with Llama and Qwen.

**Strengths:**

- The key idea that the complexity of input data varies so we should dynamically adjust sparse attention patterns and computational budget with respect to different input examples is interesting and reasonable.
- Experimental results indicate that FlexPrefill achieves comparable results with previous SOTA methods with GLM and Yi, and outperforms previous SOTA methods with a large margin with Llama and Qwen.

**Weaknesses:**

1. Figure 8 indicates that only a small portion of attention heads use Query-Aware patterns, whereas the fallback Vertical/Slash pattern is also applied in the baseline method, MInference. Given this observation, further illustration of the original attention patterns and an analysis of why FlexPrefill demonstrates significant performance gains over MInference are necessary.

**Questions:**

(a) How is the subset of queries $\hat{Q}$ selected for representative query selection in Query-Aware sparse pattern determination and Vertical-Slash head index selection? Is it randomly sampled?

(b) How was Figure 8 generated? Was it based on a specific case study or aggregated over all examples from one of the benchmarks?

(c) In Algorithm 4, the sparse index is selected globally (indicated by the flatten operation). What if perform query-wise index selection?

---

> ### Author Response · Authors · 2024-11-19
> **Title: Response to Reviewer 35xX (1/2): Response to the Weaknesses Part**
>
> **R W1: Limited Use of Query-Aware Patterns**
>
> We appreciate the insightful observation regarding the role of Query-Aware patterns and would like to elucidate FlexPrefill's advantages by focusing on two key aspects: **Pattern Determination Strategy** and **Cumulative** **Attention Mechanism**.
>
> 1. **Pattern Determination Strategy**
>
> While Query-Aware patterns may appear less frequently, their integration with Vertical-Slash patterns is crucial for enhancing model performance. Unlike MInference, which employs a fixed pattern approach, FlexPrefill dynamically selects the most appropriate pattern for each attention head, leveraging a flexible and adaptive strategy.
>
> As illustrated in **Figure 9**, our empirical analysis reveals that Query-Aware attention heads can capture more complex attention patterns, complementing the Vertical-Slash patterns without sacrificing overall model capabilities. Furthermore, the importance of Query-Aware patterns extends beyond the specific examples analyzed in **Figure 8**, which focuses on four particular inputs. In **R Q2**, we provide a more comprehensive statistical analysis, demonstrating the distribution of Query-Aware patterns across different layers and their significance as a crucial component of the model's design.
>
> 2. **Cumulative** **Attention Mechanism**
>
> FlexPrefill introduces a cumulative-attention-based index selection mechanism, which goes beyond merely choosing attention patterns. By dynamically adjusting the cumulative attention threshold (denoted as $\gamma$), FlexPrefill achieves better performance while maintaining faster inference speeds compared to MInference.
>
> As shown in **Figure 4**, the adaptability of the cumulative attention mechanism allows FlexPrefill to transition from initial underperformance to surpassing MInference with careful tuning of $\gamma$. This is reflected in the purple curve, where FlexPrefill demonstrates its ability to optimize the trade-off between performance and computational efficiency.
>
> In conclusion, this dual innovation—dynamic pattern determination and cumulative attention—positions FlexPrefill as a more adaptable and efficient alternative to MInference. By allowing for optimal trade-offs between computational cost and model effectiveness across various scenarios, FlexPrefill achieves a performance-efficiency balance that meets diverse operational requirements.

---

> ### Author Response · Authors · 2024-11-19
> **Title: Response to Reviewer 35xX (2/2): Response to the Questions Part**
>
> **R Q1: Selection method for representative query subset Q^**
>
> As detailed in our implementation section (lines 322-323), we utilize the last block_size query vectors as the representative query set $\hat{Q}$. This choice is motivated by two key factors:
>
> 1. Performance Efficacy: Our ablation studies in Table 6 demonstrate that using the last block consistently achieves better performance (89.18% avg accuracy) compared to alternative selections (e.g., middle block: 38.18% avg accuracy)
> 2. Complete Key Coverage: The last queries have visibility to all previous keys, enabling more comprehensive pattern assessment
>
>
> **R Q2: Methodology behind Figure 8 generation**
>
> Figure 8 in our paper presents an illustrative example to demonstrate how attention patterns vary across 4 different input scenarios. To provide a more comprehensive view of pattern distribution, we conducted extensive analysis across the entire RULER dataset using the Qwen2 model to obtain statistical results for Query-Aware head distributions over different layers.
>
> | Layer Range |        |        |        |        |        |        |        |
> | ----------- | ------ | ------ | ------ | ------ | ------ | ------ | ------ |
> | Layer 01-07 | 48.87% | 70.84% | 65.31% | 59.17% | 3.65%  | 18.24% | 57.65% |
> | Layer 08-14 | 14.60% | 49.75% | 21.75% | 51.75% | 3.56%  | 37.83% | 53.46% |
> | Layer 15-21 | 35.01% | 47.52% | 44.27% | 26.69% | 34.01% | 39.71% | 12.36% |
> | Layer 21-28 | 9.25%  | 10.73% | 26.89% | 9.63%  | 14.72% | 23.38% | 19.39% |
>
> Our analysis reveals that the pattern distribution shows significant layer-wise variation. Early layers (1-7) show a higher proportion of Query-Aware patterns (averaging around 46%), while deeper layers (21-28) tend to rely more on Vertical-Slash patterns (Query-Aware patterns around 16%).
>
>
> **R Q3: Global versus query-wise index selection in Algorithm 4**
>
> We conducted comprehensive experiments comparing global index selection (with flatten operation) versus query-wise selection across different models. The results demonstrate comparable performance:
>
> For average performance across all context lengths:
>
> | Ruler avg score | Llama | GLM   | Yi    | Qwen  |
> | --------------- | ----- | ----- | ----- | ----- |
> | w Flatten       | 89.18 | 89.34 | 76.32 | 70.05 |
> | wo Flatten      | 89.3  | 89.78 | 76.08 | 70.55 |
>
> For long-context scenarios (128k tokens):
>
> | Ruler 128k score | Llama | GLM   | Yi    | Qwen  |
> | ---------------- | ----- | ----- | ----- | ----- |
> | w Flatten        | 79.09 | 83.41 | 56.97 | 20.67 |
> | wo Flatten       | 78.84 | 83.65 | 57.21 | 20.91 |
>
> Given these comparable results, we opted for the global approach with flatten operation primarily for implementation efficiency. This choice simplifies the implementation of the attention mechanism while maintaining performance parity.

---

> > ### Comment · Reviewer_35xX · 2024-11-27
> >
> > My questions have been addressed. Have updated my rating.

---

### Author Response · Authors · 2024-11-22
**Response to All Reviewers**

We sincerely thank all reviewers for their thorough evaluation of our manuscript and their constructive feedback. We greatly appreciate the time and effort they have invested in reviewing our work.
We are deeply grateful to all reviewers for their careful reading of our manuscript and their detailed, constructive feedback. Their insights have helped us improve the quality and clarity of our paper significantly.

**We have carefully addressed all the suggestions and feedback in our revised manuscript.** The major changes are highlighted in red in the updated paper, including:
- Additional experimental results as requested
- Improved clarity in methodology descriptions
- Fixed typographical errors

We believe these revisions have strengthened our paper while maintaining its core contributions and insights.
Once again, we thank the reviewers for their valuable feedback that has helped us improve our manuscript. We look forward to their continued engagement with our work.

Sincerely,

The Authors

---

### Meta-Review · Area_Chair_Ldfh · 2024-12-20

**Metareview:**

This paper introduces FlexPrefill, a novel sparse attention mechanism for efficient long-sequence inference in large language models (LLMs). It tackles the computational challenges associated with long-sequence processing by dynamically adjusting sparse attention patterns and computational budgets based on input characteristics. Experimental results show improvements in both speed and accuracy compared to existing methods.

The reviewers reach the consensus that this paper is technical sound with solid contributions. The experiments are comprehensive, well supporting its claims. There are also questions regarding the clarity and additional experiments, which are addressed during the author-reviewer discussion. I agree with the reviewers and would recommend acceptance of this work.

**Additional Comments On Reviewer Discussion:**

There are questions regarding the clarity and additional experiments from the reviews. Most of them are addressed during the author-reviewer discussion.

---

### Decision · Program_Chairs · 2025-01-22

Accept (Oral)